Microbiology
SPECTRUM
# Statistical Evaluation of Metaproteomics and 16S rRNA Amplicon Sequencing Techniques for Study of Gut Microbiota Establishment in Infants with Cystic Fibrosis

Claudia Saralegui,[a,b,c] Carmen García-Durán,[d] Eduardo Romeu,[e] María Luisa Hernáez-Sánchez,[e] Ainhize Maruri,[a,b,c] Natalia Bastón-Paz,[a,b,c] Adelaida Lamas,[c,f] Saioa Vicente,[c,f] Estela Pérez-Ruiz,[g] Isabel Delgado,[h] Carmen Luna-Paredes,[i] Juan de Dios Caballero,[a,b,c] Javier Zamora,[j] ⓘ Lucía Monteoliva,[d,e] Concepción Gil,[d,e] ⓘ Rosa del Campo[a,b,c,k]

aServicio de Microbiología, Hospital Universitario Ramón y Cajal and Instituto Ramón y Cajal de Investigación Sanitaria, Madrid, Spain
bCIBERINFEC, Madrid, Spain
cUnidad de Fibrosis Quística, Hospital Universitario Ramón y Cajal, Madrid, Spain
dDepartamento de Microbiología y Parasitología, Universidad Complutense de Madrid and Instituto Ramón y Cajal de Investigación Sanitaria, Madrid, Spain
eUnidad de Proteómica, Universidad Complutense de Madrid, Madrid, Spain
fServicio de Pediatría, Hospital Universitario Ramón y Cajal and Instituto Ramón y Cajal de Investigación Sanitaria, Madrid, Spain
gUnidad de Fibrosis Quística, Hospital Regional Universitario de Málaga, Málaga, Spain
hUnidad de Fibrosis Quística, Hospital Virgen del Rocío, Seville, Spain
iSección de Neumología y Alergia Infantil, Unidad Multidisciplinar Fibrosis Quística, Hospital Doce de Octubre, Madrid, Spain
jUnidad de Bioestadística, Hospital Universitario Ramón y Cajal and Instituto Ramón y Cajal de Investigación Sanitaria and CIBERESP, Madrid, Spain
kUniversidad Alfonso X El Sabio, Madrid, Spain

Claudia Saralegui and Carmen García-Durán contributed equally to this work. The order of the authors was decided by their contributions.

**ABSTRACT** Newborn screening for cystic fibrosis (CF) can identify affected but asymptomatic infants. The selection of omic technique for gut microbiota study is crucial due to both the small amount of feces available and the low microorganism load. Our aims were to compare the agreement between 16S rRNA amplicon sequencing and metaproteomics by a robust statistical analysis, including both presence and abundance of taxa, to describe the sequential establishment of the gut microbiota during the first year of life in a small size sample (8 infants and 28 fecal samples). The taxonomic assignations by the two techniques were similar, whereas certain discrepancies were observed in the abundance detection, mostly the lower predicted relative abundance of *Bifidobacterium* and the higher predicted relative abundance of certain *Firmicutes* and *Proteobacteria* by amplicon sequencing. During the first months of life, the CF gut microbiota is characterized by a significant enrichment of *Ruminococcus gnavus*, the expression of certain virulent bacterial traits, and the detection of human inflammation-related proteins. Metaproteomics provides information on composition and functionality, as well as data on host-microbiome interactions. Its strength is the identification and quantification of *Actinobacteria* and certain classes of *Firmicutes*, but alpha diversity indices are not comparable to those of amplicon sequencing. Both techniques detected an aberrant microbiota in our small cohort of infants with CF during their first year of life, dominated by the enrichment of *R. gnavus* within a human inflammatory environment.

**IMPORTANCE** In recent years, some techniques have been incorporated for the study of microbial ecosystems, being 16S rRNA gene sequencing being the most widely used. Metaproteomics provides the advantage of identifying the interaction between microorganisms and human cells, but the available databases are less extensive as well as imprecise. Few studies compare the statistical differences between the two techniques to define the composition of an ecosystem. Our work shows that the two methods are comparable in terms of microorganism identification but provide different results in alpha diversity analysis. On the other hand, we have studied newborns with cystic fibrosis, for whom we have

Address correspondence to Concepción Gil, conchagil@ucm.es, or Rosa del Campo, rosa.campo@salud.madrid.org.

The authors declare a conflict of interest. Rosa del Campo is recipient of a Vertex grant. The remaining authors have no conflict of interest.

[This article was published on 18 October 2022 with information missing from Acknowledgments. This information was added in the current version, posted on 15 November 2022.]

described the establishment of an intestinal ecosystem marked by the inflammatory response of the host and the enrichment of *Ruminococcus gnavus*.

**KEYWORDS** amplicon sequencing, metaproteomics, cystic fibrosis, gut microbiota establishment, *Ruminococcus gnavus*, Bland-Altman test

The establishment of the microbiota at birth affects human health during the rest of life (1). In cystic fibrosis (CF), a genetic disease diagnosed by neonatal screening, the colonization of the lung airways by pathogenic bacteria impacts survival (2). The life expectancy of individuals with CF has increased considerably in recent years, and comorbidities beyond those of the respiratory tract have been revealed, mostly related to the digestive tract (3). The metabolic activity of the gut microbiota has a direct impact on nutritional status, a relevant aspect in CF since most of the patients are malnourished, and proper nutrition has a major effect on their infectious lung disease (4).

In CF, the gut microbiota is conditioned by the following factors: transmembrane regulator dysfunction, which creates a thick mucus layer; the acidic environment caused by impaired bicarbonate secretion; maldigestion; malabsorption; steatorrhea due to pancreatic insufficiency; and the high intake of antibiotics for bacterial infections during pulmonary exacerbations (5, 6). Significant differences in the microbiota's bacterial composition and alpha diversity have been reported compared with that in healthy infants (7–9).

Currently, 16S rRNA amplicon sequencing (AS) is the most extensive, rapid, and cost-effective technique for characterizing complex bacterial communities, such as the gut microbiota (10). The technique's main weakness results from the variable affinity of the primers for flanking regions; consequently, a higher predicted relative abundance has been found for certain genera, whereas others, mainly *Bifidobacterium*, are consistently reported to have a lower predicted relative abundance than other methods (11). Even with optimized primers, amplicon sequencing often provides deficient taxonomic assignment at the genus and species levels due to the short length of the amplicons. On the other hand, metaproteomics applies high-resolution mass spectrometry (MS) for the detection and quantification of proteins, allowing both taxonomic identification and functional characterization of all present microorganisms, not just bacteria. The recent introduction of updated bioinformatics tools, such as MetaLab, has considerably improved data analysis (12); for lesser-represented species, however, the analysis remains inaccurate (13). Available data suggest that only 10% to 20% of the total protein content of a clinical sample is identified, corresponding to only one-third of the entire microbiota (14). Additional strategies have recently been suggested as prefractionation steps to circumvent instrument sensitivity problems and access low-abundance proteins of biological importance (15). The strengths of metaproteomics include a high resolution in bacterial species identification, the overview of functional and metabolic activities of microorganisms, and, most notably, the detection of host proteins, expanding the knowledge of host-microbiome interactions (16, 17), which is particularly interesting in CF (18). However, metaproteomics also has relevant limitations (19): standardization of the technique is needed to make it reproducible, and integration of meta-omics platforms is necessary for in-depth characterization. Moreover, although its cost has been reduced, it is still far from those for amplicon sequencing.

The combination of amplicon sequencing and metaproteomics techniques provides a wider characterization of microbial communities and their interaction with the host (20–22), and the integration of the results could support accurate predictions (23). However, few studies have included a statistical analysis comparing the results by the two methodologies, especially for challenging samples such as those from neonates, characterized by limited sample quantity and potentially low bacterial load. The aims of this study were to perform a robust statistical analysis on the concordance of amplicon sequencing and metaproteomics to study and describe the sequential establishment of the gut microbiota of a small representative cohort of infants with CF.

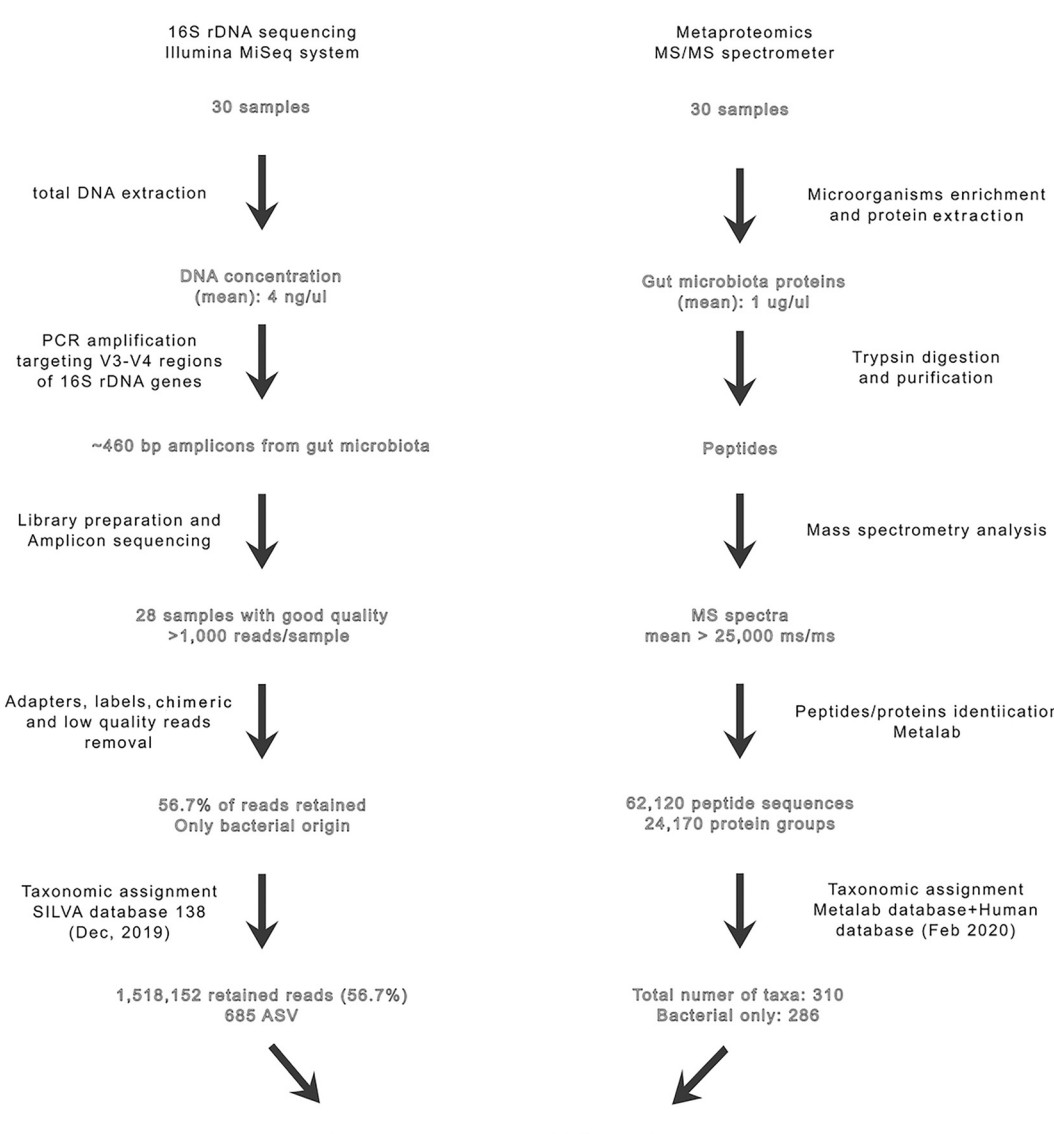

**FIG 1** Workflow and main results obtained from the two methodologies used in this study.

## RESULTS

**Statistical concordance of the results of amplicon sequencing and metaproteomics.**
We analyzed fecal samples ($n = 28$) using both techniques for bacterial identification
(presence/absence) and quantification (abundance) and then respectively applied the
McNemar and Wilcoxon tests to detect significant discrepancies (Fig. 1; see also "Amplicon
sequencing data" and "Metaproteomics" in the supplemental material). Globally, 685
amplicon sequence variants (ASVs) (9 phyla, 13 classes, and 130 genera) were identi-
fied by amplicon sequencing (AS), and 310 taxa (8 phyla, 17 classes, and 65 genera) were
identified by metaproteomics.

Taxa detected exclusively by a single method accounted for less than 10% of the total
bacterial composition, being the genus level the one with most of the unique bacteria
(Fig. 2). Minor statistical discrepancies were observed in taxon presence/absence by McNemar
test (Table 1), most of which were related to the genus level (Fig. 2, Table 2, and "Common
taxonomy" in the supplemental material). Furthermore, we detected certain nominal inconsis-
tencies that should be accounted for related to the taxonomic classification by each database,
such as the inclusion of the *Tenericutes* phylum from metaproteomics in the *Firmicutes* phylum

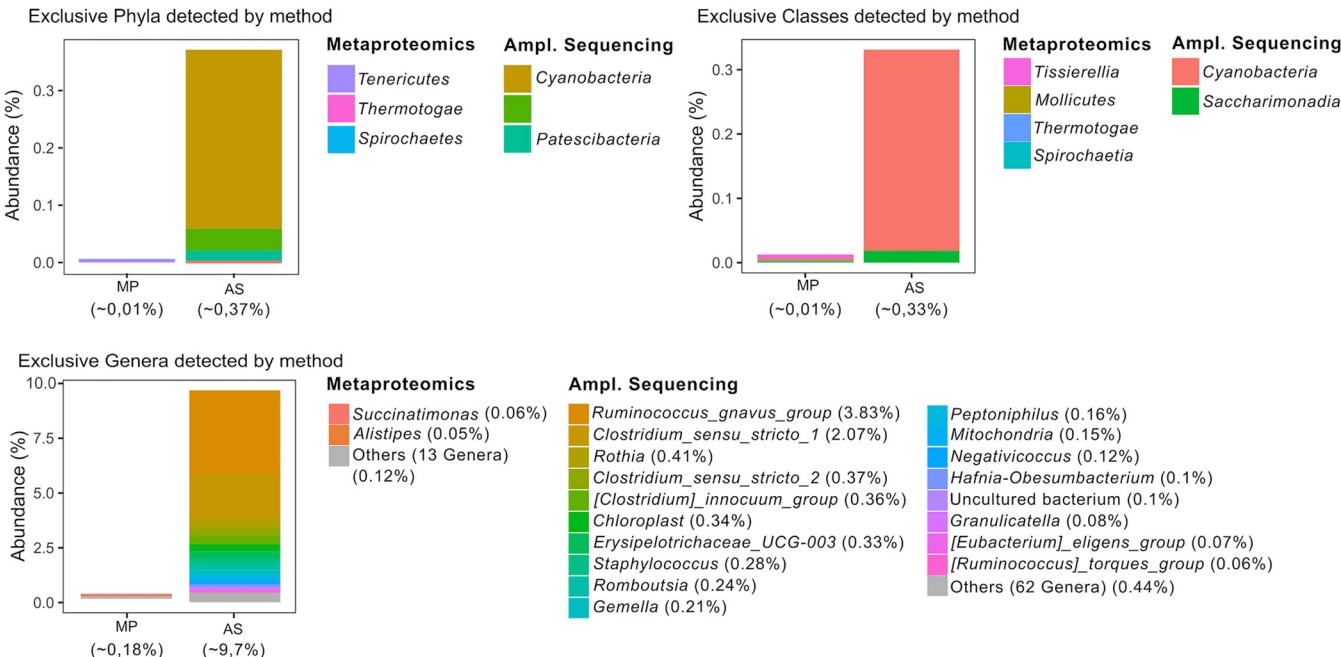

**FIG 2** Bacterial phyla, classes, and genera uniquely identified by each of the methodologies. AS, amplicon sequencing; MP, metaproteomics. White spaces correspond to taxa correctly identified by both methods.

according to amplicon sequencing and the recent taxonomic reclassification of *Proteobacteria* classes (Table 2). Most of the reclassified taxa were at higher taxonomic levels.

Figure 3 summarizes the results of the compositional analysis from the top common taxa. Wilcoxon signed-rank paired tests showed significant differences in the relative abundances of the following taxa: the phylum *Actinobacteriota*, the classes *Bacilli* and *Actinobacteria*, and the genera *Blautia*, *Haemophilus*, *Escherichia*, *Enterococcus*, *Streptococcus*, and *Bifidobacterium* (Fig. 3). These top genera were separately explored to consider within-subject effect across the time series (Fig. 4A). The only genera that had a differential composition across time according to AS were *Ruminococcus* ($P = 0.022$) and *Eubacterium* ($P = 0.007$) (Fig. 4B, left). According to metaproteomics, the genera that have changed over time are *Fusobacterium*

**TABLE 1** Contingency table for McNemar test accounting for the number of samples in which different bacteria (class and genus levels) were or were not identified by one, both, or neither of the technologies

| Organism(s) | No. of samples detected by: | | | | |
| --- | --- | --- | --- | --- | --- |
| | Amplicon sequencing only | Metaproteomics only | Both | Neither | *P* value[a] |
| Class | | | | | |
| *Alphaproteobacteria* | 8 | 1 | 9 | 10 | 0.02 |
| | | | | | |
| Genus | | | | | |
| *Eggerthella* | 12 | 0 | 1 | 15 | 0.001 |
| *Anaerococcus* | 11 | 0 | 1 | 16 | 0.002 |
| *Haemophilus* | 15 | 2 | 5 | 6 | 0.003 |
| *Coprococcus* | 1 | 10 | 1 | 16 | 0.007 |
| *Lachnoclostridium* | 0 | 9 | 2 | 17 | 0.007 |
| *Roseburia* | 0 | 9 | 2 | 17 | 0.007 |
| *Eubacterium* | 1 | 14 | 6 | 7 | 0.01 |
| *Enterobacter* | 10 | 1 | 3 | 14 | 0.01 |
| *Enterococcus* | 10 | 1 | 14 | 3 | 0.01 |
| *Escherichia* | 11 | 1 | 14 | 2 | 0.01 |
| *Actinomyces* | 14 | 4 | 6 | 4 | 0.03 |
| *Collinsella* | 8 | 1 | 5 | 14 | 0.04 |

[a]Significance level < 0.05.

**TABLE 2** Discrepancies in nomenclature or taxonomic classification at different ranks (phylum/class/order/family/genus) by amplicon sequencing and metaproteomics[a]

| Taxa identified by amplicon sequencing | Taxa identified by metaproteomics |
|---|---|
| *Campilobacterota/Campylobacteria/Campylobacterales* | *Proteobacteria[a]/Epsilonproteobacteria/Campylobacterales* |
| *Firmicutes/Bacilli/Mycoplasmatales* | *Tenericutes/Mollicutes/Mycoplasmatales* |
| *Actinobacteriota* | *Actinobacteria* |
| *Bacteroidota* | *Bacteroidetes* |
| *. . ./Bacilli/Erysipelotrichales* | *. . ./Erysipelotrichia/Erysipelotrichales* |
| *. . ./Gammaproteobacteria/Burkholderiales* | *. . ./Betaproteobacteria/Burkholderiales* |
| *Desulfobacterota/Desulfovibrionia/Desulfovibrionales* | *Proteobacteria/Deltaproteobacteria/Desulfovibrionales* |
| *. . ./Clostridia/Peptostreptococcales-Tissierellales* | *. . ./Tissierellia/Tissierellales* |
| *. . ./. . ./. . ./. . ./Escherichia-Shigella* | *. . ./. . ./. . ./. . ./Escherichia* |
| *. . ./. . ./Lachnospirales/Lachnospiraceae/Ruminococcus_gnavus_group* | *. . ./. . ./Clostridiales/Ruminococcaceae/Ruminococcus* |
| *. . ./. . ./Lachnospirales/Lachnospiraceae/Ruminococcus_torques_group* | |

[a]Names of taxonomic ranks that are identical are expressed with ". . ./." The taxonomic assignment was made in 2019 before the latest nominal changes in the bacterial phyla. *Actinobacteriota* is now *Actinomycetota*, *Firmicutes* is *Bacillota*, and *Proteobacteria* is now *Pseudomonadota*.

($P = 0.02$), *Haemophilus* ($P = 0.029$) and *Veillonella* ($P = 0.03$) (Fig. 4B, right). The two methods showed similar trends for most genera, except for *Bifidobacterium*, *Escherichia*, *Enterococcus*, and *Ruminococcus*, which almost reach statistical significance ($P < 0.05$ but with a false-discovery rate [FDR] of 0.13). *Bifidobacterium* was the genus with the highest distance between methods (Fig. 4C to E). It is worth mentioning that significance could be affected by the small sample size and that sometimes changes are mostly detected for the lowest-abundance genera.

Alpha diversity metrics were significantly different between methodologies (Wilcoxon signed-rank paired test, $P < 0.05$). The methods also showed differences in temporal trends (Fig. 5), with only amplicon sequencing showing an increasing trend for the Shannon index in the latest samples. It must be mentioned that alpha diversity metrics calculations did not work well with metaproteomics data, as phyloseq software warned that there were no singletons in metaproteomics data frames. Calculating Shannon indices from rank-collapsed data and label free quantifications (LFQ) intensities as the ones that were available in metaproteomics workflow output is not common and should be considered for further works. Amplicon sequencing remained the best approach to calculate diversity metrics such as the aforementioned.

**Functional assignment by metaproteomics.** Functional information on the detected proteins was obtained by MetaLab software, using the sum of the peptide intensities for each protein (total intensity of the protein). Both the number and intensity of the proteins in the initial (5,718 and 4.86e12, respectively) and the early-CF (7,029 and 4.70e12, respectively) fecal samples were comparable. MetaLab assigned the functional profile by matching proteins with the Cluster of Orthologous Groups (COG) database with a 97% successful assignment rate for name/category. A previous process of normalization of the protein intensities (intensity of the function/sum of the intensities in that sample) was performed, and 2-way analysis of variance detected 14 COGs (6 from the initial and 8 from the early-CF samples) with statistical relevance (Table 3). However, and despite the small sample size of our work, there were no significant differences in the general functional profiles in the enrichment analysis, of which "energy production and conversion" (C), "carbohydrate transport and metabolism" (G), and "translation, ribosomal structure, and biogenesis" (J) were the COG categories with greater intensity (Fig. 6A).

The correlation of each bacterial taxon and its function was also performed with iMetaLab, using the iMetaShiny apps. *Bifidobacterium* spp. were the most abundant taxa and their proteins remained abundant in the early-CF samples, in which most of the functional pathways were promoted by *Ruminococcus gnavus* (Fig. 6B). A significant enrichment of *R. gnavus* proteins was observed from initial to early-CF times (2 versus 69 proteins), most of them related to translational processes (Fig. 6C). In contrast, the functions associated with *Clostridium perfringens* and *Blautia hansenii* showed a statistically significant decrease in the early-CF samples.

Metaproteomics also allowed the identification of 293 human proteins: 85 exclusively

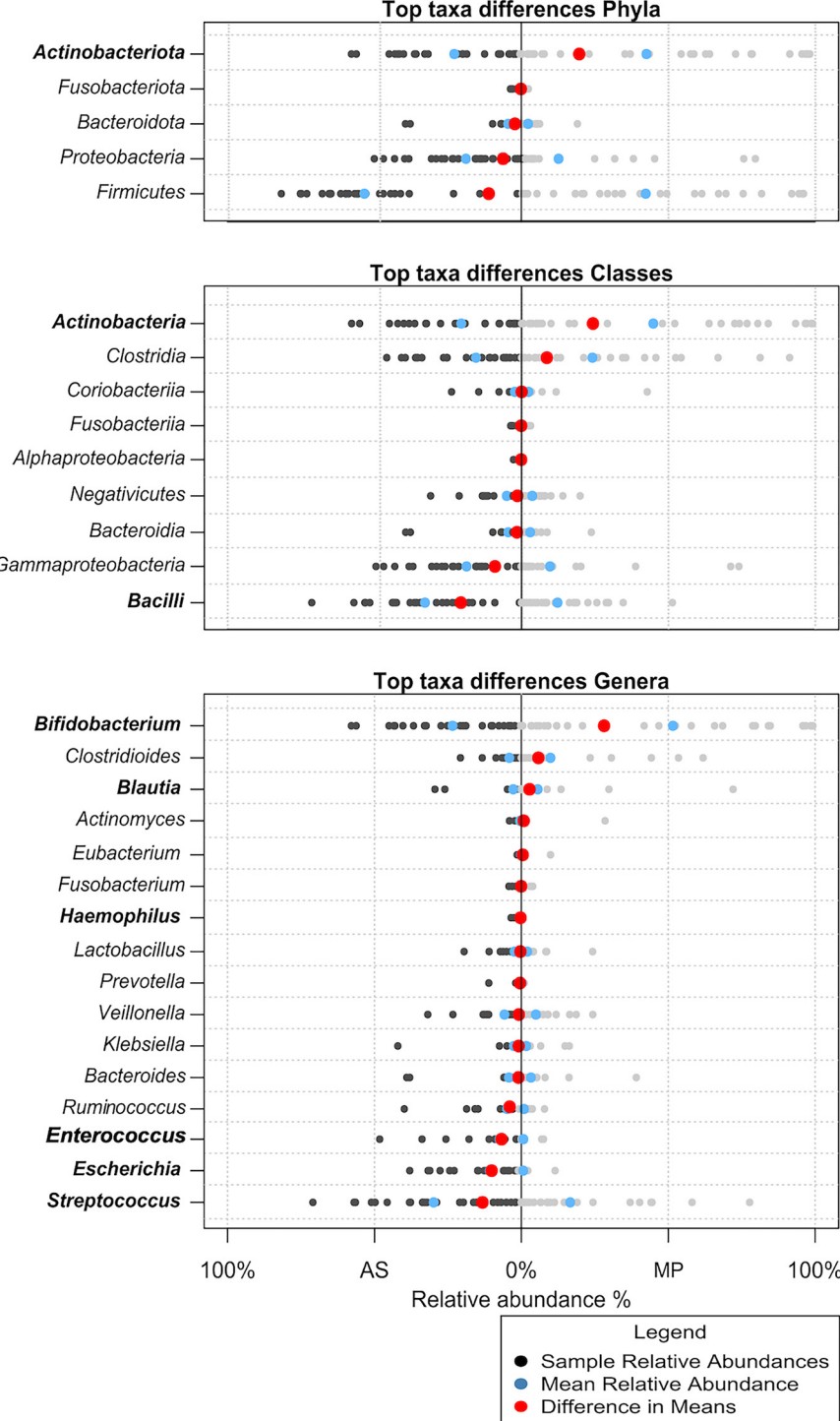

**FIG 3** Relative abundances of the main taxa detected by both methodologies. Taxa in bold are those for which the paired Wilcoxon test result was significant ($P < 0.05$).

present in initial samples, 43 exclusively in early-CF samples, and 165 in both samples ("Human proteins" in the supplemental material). Six of the 165 shared proteins had significantly differential expression levels (Table 4). A STRING analysis (STRING version 11.5) was performed with the 3 groups of human proteins (initial, early CF, and shared proteins), including for the first 2 groups those proteins exclusively identified and with a significantly higher abundance in each of the samples (Fig. 7). The analysis of the proteins shared by the two groups

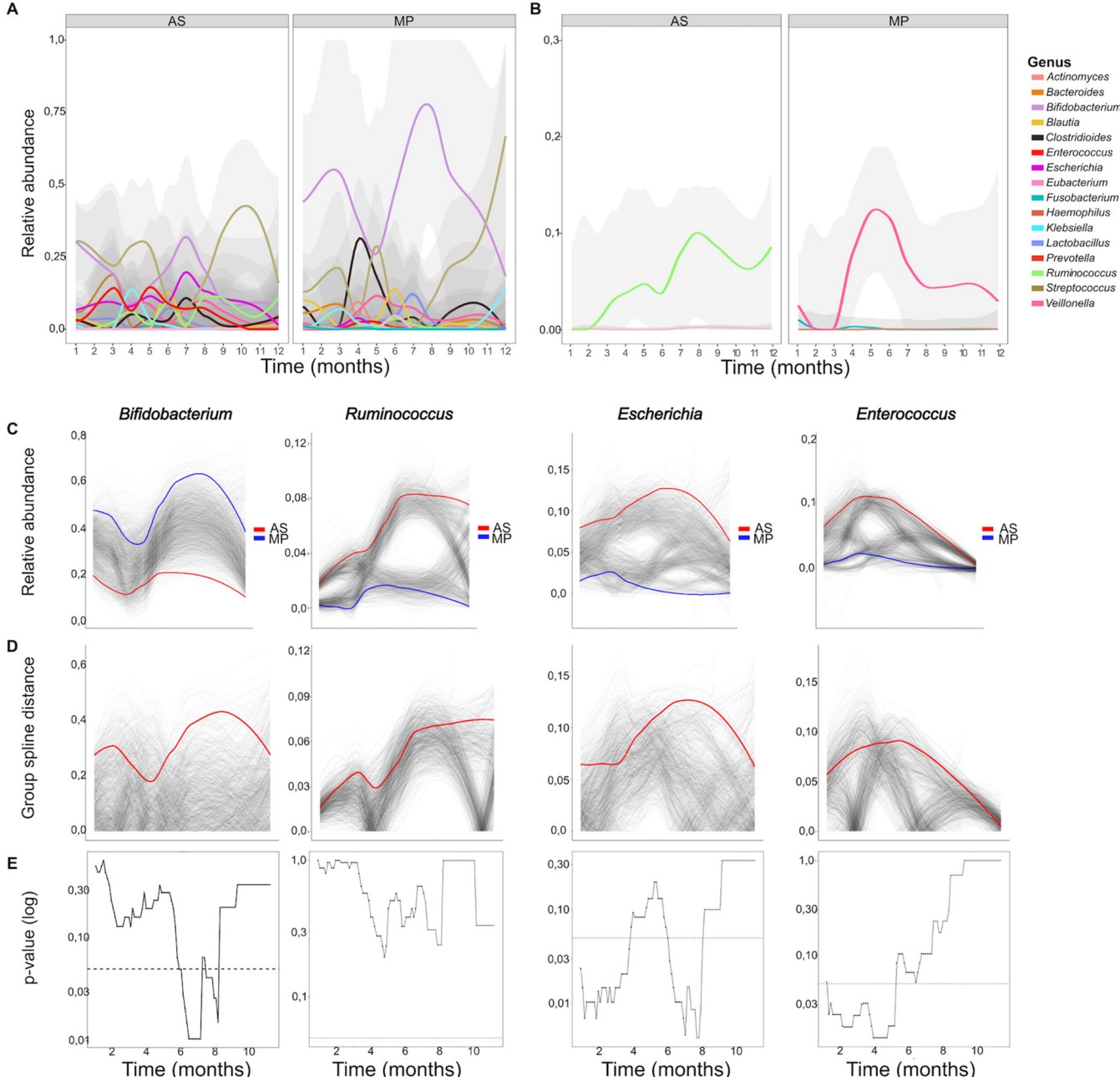

**FIG 4** Compositional change of the major genera in each of the methodologies. (A) Evolution of the relative abundance of the main bacterial genera ($n$ = 16). (B) Evolution of bacterial genera with a significant change over time detected by AS (*Ruminococcus* and *Eubacterium*) and by MP (*Fusobacterium*, *Haemophilus*, and *Veillonella*). (C) Graphs obtained by the permuspliner function (999 permutations) showing the temporal evolution of relative abundance in those genera with significant differences between the two methodologies. (D) Plots of distances between both methodologies. The difference (solid red line) is not significant if it is not above 95% of the permuted values (translucent gray). (E) Plots obtained by sliding spliner showing the *P* value at each specified interval (shown with 100 intervals by default). The dotted line indicates a *P* value of 0.05. At certain intervals, the differences became significant.

showed a significant protein-protein interaction ($1.87 \times 10^{-16}$). Shared proteins were mainly related to neutrophil-mediated immunity (42 proteins; $P = 1.87 \times 10^{-25}$), such as calprotectin (S100-A9 and S100-A8), neutrophil elastase (ELANE), cathepsin G, and myeloperoxidase, or related to the intestinal barrier, such as mucins (MUC2, MUC4, and MUC13) or proteins present in cell junctions (40 proteins; $P = 1.24 \times 10^{-5}$) (Fig. 7A). The initial samples contained significantly more proteins associated with chylomicron assembly and the immune system than the early-CF samples (Fig. 7B), in which a cluster of muscle proteins stood out (Fig. 7C).

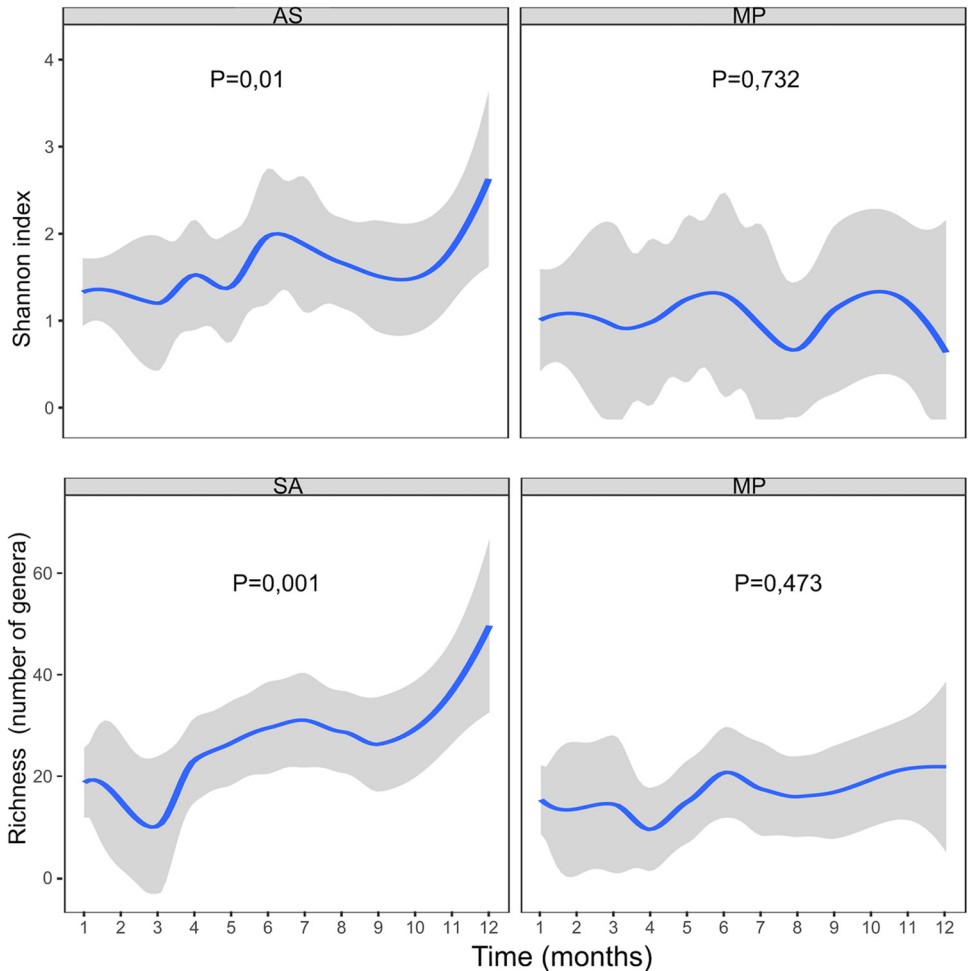

**FIG 5** Evolution of alpha diversity (measured by Shannon index and richness) detected by one method and the other. The *P* value was obtained with the trendyspliner spliner function (shown with 100 intervals by default), which evaluates the differences over time for each group of samples, separately.

The names of the proteins belonging to each function are listed in "Microbial data" in the supplemental material.

## DISCUSSION

A major clinical feature of CF are digestive comorbidities, such as steatorrhea, maldigestion/malabsorption, and pancreatic insufficiency, which also affect the quality of life and nutritional status of these individuals. A new classification of CF genetic mutations has been published (24), differentiating between high and low risk according to their clinical manifestations. Seven out of the 8 infants of this study carried high-risk mutations. The CF environment strongly conditions the gut microbiota composition (25), also contributing to its malfunction, even at the age of 6 weeks (9). Even though amplicon sequencing is the most widely used technique for determining the composition of gut microbiota, there is a growing demand for techniques that determine the global functionality of the ecosystem, such as metaproteomics.

The taxonomical incongruities detected were attributed to paraphyletic groups, which are continuously reclassified according to the latest advances in phylogeny. In this sense, SILVA 138 is a more accurate database with which to establish phylogenetic relationships, because protein databases are less frequently updated. By the time this paper is published, certain phylum names will have changed: *Firmicutes* is now *Bacillota*, *Proteobacteria* is now *Pseudomonadota*, and *Actinobacteriota* is currently named *Actinomycetota*.

**TABLE 3** Functional categories of bacterial proteins with significantly different values between the initial group of samples and the early-CF group

| COG name[a] | Mean difference | P value | COG category |
|---|---|---|---|
| Formyltetrahydrofolate synthetase (2,402) | 2.64 | <0.0001 | Nucleotide transport and metabolism |
| ABC-type phosphate transport system, periplasmic component (2,605) | 1.20 | <0.0001 | Inorganic ion transport and metabolism |
| Superfamily II DNA and RNA helicase (3,843) | 1.25 | <0.0001 | Replication, recombination, and repair |
| Translation elongation factor EF-Tu, a GTPase (2,272) | 0.74 | 0.0009 | Translation, ribosomal structure, and biogenesis |
| $F_oF_1$-type ATP synthase, delta subunit (1,785) | 0.65 | 0.01 | Energy production and conversion |
| Pyruvate-formate lyase (4,465) | 0.63 | 0.03 | Energy production and conversion |
| Chaperonin GroEL (HSP60 family) (2,675) | −2.06 | <0.0001 | Posttranslational modification, protein turnover, and chaperones |
| Glyceraldehyde-3-phosphate dehydrogenase/erythrose-4-phosphate dehydrogenase (2,333) | −1.03 | <0.0001 | Carbohydrate transport and metabolism |
| Carboxysome shell and ethanolamine utilization microcompartment protein CcmL/EutN (994) | −0.98 | <0.0001 | Secondary-metabolite biosynthesis, transport, and catabolism |
| Phosphoenolpyruvate-protein kinase (PTS system component in bacteria) (4,879) | −0.96 | <0.0001 | Carbohydrate transport and metabolism |
| Alcohol dehydrogenase, class IV (6,245) | −0.94 | <0.0001 | Energy production and conversion |
| Rubrerythrin (4,121) | −0.86 | <0.0001 | Energy production and conversion |
| Threonine dehydrogenase or related Zn-dependent dehydrogenase (5,594) | −0.79 | <0.0001 | General function prediction only |
| ABC-type sugar transport system, periplasmic component, contains N-terminal Xre family HTH domain (4,787) | −0.62 | 0.03 | Carbohydrate transport and metabolism |

[a]PTS, phosphotransferase system; HTH, helix-turn-helix.

Curiously, metaproteomics detected proteins attributed to *Streptococcus thermophilus*, whereas when using amplicon sequencing, the proteins appeared to correspond to *Streptococcus salivarius*. Poor resolution of streptococcal species using the V3-V4 regions of the 16S rRNA gene has been previously reported (26), whereas the accuracy of metaproteomics in bacterial species identification has been demonstrated (12). Another amplicon sequencing limitation is the detection of what is known as sporobiota (27), most of the endosporulating species belonging to *Firmicutes*. Spores are underrepresented because of their ability to resist DNA extraction techniques. Other bacterial phyla, such as *Actinobacteria* (particularly *Bifidobacterium*), have genomes with a high GC content, and this likewise contributes to their underestimation by sequencing techniques.

Metaproteomics has also failed to detect or properly estimate certain bacterial groups typically found in the human gut microbiota due to their resistance to cell lysis, especially Gram-positive bacteria (28). Moreover, certain microbial components and detergents used for cell lysis can affect postenzymatic digestion (14). Our workflow includes optimized steps, such as serial centrifugations for microbial preenrichment, sonication with bead beating, and lysis with sodium dodecyl sulfate (SDS)-based buffers. While our method improves the number of peptide and protein identifications and detection of the relative abundance of *Actinobacteria*, other protocols have shown a higher identification of proteins from other taxonomic groups, such as *Proteobacteria*, but a lower number of peptide and protein identifications (29). Metaproteomic analysis presents several bioinformatics challenges, the most common being adequate peptide identification, taxonomic assignment, and the quality of functional annotation (14).

The statistical analysis of our results demonstrated equivalent bacterial taxonomic assignations by both techniques, while for abundance detection, some discrepancies were found. The lower predicted relative abundance of *Bifidobacterium* by amplicon sequencing has been repeatedly reported, as well as the higher predicted relative abundance we detected for some *Firmicutes* and *Proteobacteria* taxa in comparison to other methodologies (30).

The small size of our cohort limits comparison with other studies. Alpha diversity indices of our samples did not reach those observed in other CF cohorts or in the control cohort of Antosca et al. (9), although other work showed lower values in CF patients than in healthy controls (31). We conclude that it is not possible to compare alpha diversity indices between metaproteomics and amplicon sequencing with the approach we followed in this study,

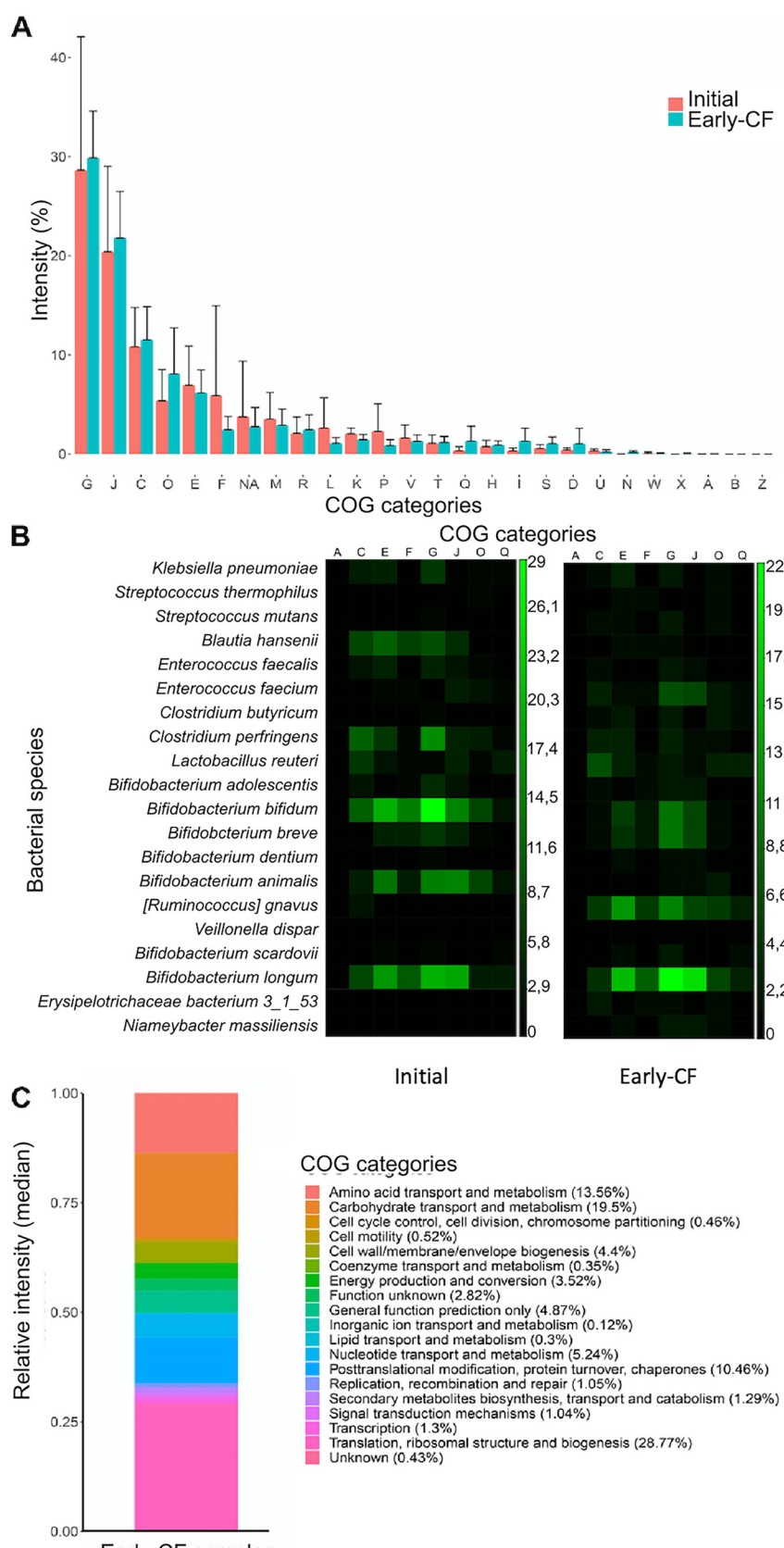

**FIG 6** Functional analysis of bacterial proteins from this cohort, comparing the group of initial and final samples (early CF). (A) Relative intensity of each of the COG functional categories in each group of

**TABLE 4** Human proteins enriched in each group of the cohort

| Higher expression in initial sample | Higher expression in early-CF sample | P value |
|---|---|---|
| Lactotransferrin | | <0.0001 |
| Myeloperoxidase | | <0.0001 |
| | Actin cytoplasmic 2 | <0.0001 |
| | Chymotrypsin-C | <0.0001 |
| | Carboxypeptidase A1 | 0.0409 |
| | Myosin-1 | <0.0001 |

although comparison between samples processed with identical proteomic techniques could be possible. In amplicon sequencing, several ASVs (each with a certain number of reads, typically on the order of 1 to $10^4$) may correspond to the same bacterial taxon. In metaproteomics, on the other hand, the sum of intensities (on the order of $10^6$ to $10^{11}$) of all peptides corresponding to the same taxon is evaluated to calculate the taxon's relative abundance in every sample. This makes the calculation of metrics such as the Shannon index, which considers the proportion of each bacterium, and their comparison between techniques not entirely reliable; nonetheless, species richness (number of different species) could still be used.

Fecal proteomics for adults with CF has pointed to the dominance of *R. gnavus*, *Enterobacteriaceae*, and *Clostridia* species, combined with decreased butyrate producers, such as *Faecalibacterium prausnitzii* (18). We detected similar compositional changes in a very reduced cohort of 8 infants followed for a year right after being diagnosed with CF. *R. gnavus* has been associated with several human gastrointestinal diseases and is a mucolytic bacterium capable of degrading the human colonic mucin MUC2, which is a major form of secretory glycosylated mucin coating the epithelia of the intestine and therefore has a close relationship with the mucus layer (32, 33). It has been suggested that the bacterium can penetrate the mucus layer and lead to an activation of the immune system and inflammatory responses (34, 35). Nevertheless, the involvement of this species in allergic or autoimmune diseases has been also questioned (32, 36).

Drastic changes in the bacterial functional profiles of the initial and early-CF samples were not observed; the most abundant proteins in the initial samples were a formyltetrahydrofolate synthetase and a pyruvate-formate lyase previously related to biofilm upregulation (37). In the early-CF samples, the most relevant enrichment was that of the chaperonin GroEL, a stress response protein whose main role is protein folding, which has recently been observed to have roles in virulence and pathogenesis, such as bacterial adherence and immune invasion (38).

One of the most interesting advantages of metaproteomics is the detection of human proteins. In our samples, several proteins associated with the maintenance of the intestinal epithelium and the immune response were detected. Lactotransferrin, an iron sequestration protein abundant in human breast milk; and myeloperoxidase, regulator of inflammation in granulocytes of neutrophils, were significantly enriched in the initial samples (39, 40). All our results point to an altered establishment of the

**FIG 6** Legend (Continued)

samples. (B) Correlation between the most enriched functional categories and the bacterial species to which they are assigned. The color gradient indicates higher relative intensity (light green). (C) Proportions of COG functional categories of proteins assigned to *R. gnavus*. COG category nomenclature: A, RNA processing and modification; B, chromatin structure and dynamics; C, energy production and conversion; D, cell cycle control, cell division, and chromosome partitioning; E, amino acid transport and metabolism; F, nucleotide transport and metabolism; G, carbohydrate transport and metabolism; H, coenzyme transport and metabolism; I, lipid transport and metabolism; J, translation, ribosomal structure, and biogenesis; K, transcription; L, replication, recombination, and repair; M, cell wall/membrane/envelope biogenesis; N, cell motility; O, posttranslational modification, protein turnover, and chaperones; P, inorganic ion transport and metabolism; Q, secondary-metabolite biosynthesis, transport, and catabolism; R, general function prediction only; S, function unknown; T, signal transduction mechanisms; U, intracellular trafficking, secretion, and vesicular transport; V, defense mechanisms; W, extracellular structures; X, mobilome: prophages and transposons; Z, cytoskeleton; NA, not assigned to any category.

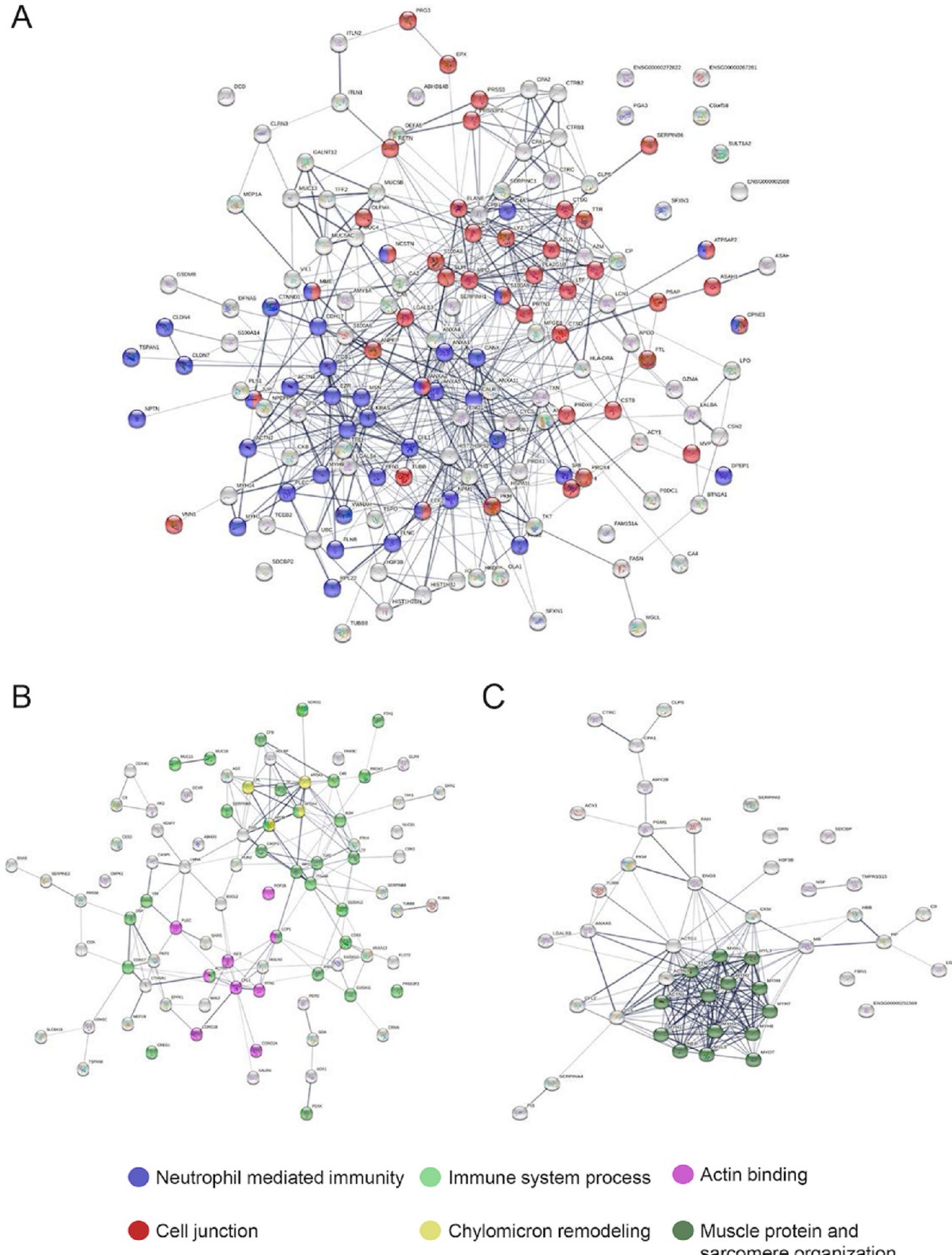

**FIG 7** Human proteins found interacting with the gut microbiome of the cohort. (A) Group of proteins unique to the initial samples; (B) group of proteins unique to the early-CF samples; (C) proteins shared by both groups.

**TABLE 5** Relevant clinical characteristics of infants and sampling[a]

| Infant | Sex | Pancreatic insufficiency | Delivery mode | Birth feeding | CF mutation[b] | Sampling at mo (no. of sample) | | | | | | | | | | | |
|---|---|---|---|---|---|---|---|---|---|---|---|---|---|---|---|---|---|
| | | | | | | 1 | 2 | 3 | 4 | 5 | 6 | 7 | 8 | 9 | 10 | 11 | 12 |
| 1 | Female | Yes | Vaginal | Formula | G542X/R1162X | 1 | 2 | | | | | | 3 | | | | |
| 2 | Male | Yes | Caesarean section | Formula | F508del/Y1092X | 4 | | | | 5 | | 6 | | | | | |
| 3 | Female | Yes | Vaginal | Exc.[c] breast milk | F508del/F508del | 7 | | | 8 | | | | | | 9 | | 10 |
| 4 | Male | No | Vaginal | Exc. breast milk | F508del/3272-26A->G | 11 | | 12 | | | 13 | | 14 | | | | |
| 5 | Male | Yes | Vaginal | Exc. breast milk | F508del/F508del | 15 | | | 16 | | | | 17 | | | | |
| 6 | Male | Yes | Vaginal | Formula | F508del/N1303K | 18 | | | | | | 19 | | 20 | | | |
| 7 | Female | Yes | Vaginal | Formula | F508del/F508del | 21 | | | 22 | 23 | | 24 | | | | | |
| 8 | Female | Yes | Vaginal | Formula | F508del/R1162X | 25 | | 26 | | 27 | | | | | | 28 | |

[a]All samples (n = 28) were included in the methodological comparative analysis, whereas samples in gray (n = 16; 8 initial samples and 8 early-CF samples) were considered to analyze gut microbiota and human proteomic content in this cohort.
[b]All CFTR alleles except 3272-26A>G are classified as high risk, and therefore, all infants had a high-risk CF genotype except infant 4.
[c]Exc., exclusively breast milk. At endpoint, all infants had included solid food in their feeding status.

gut microbiota in infants with CF, with a dominance of *R. gnavus* within a human inflammatory state.

The main limitation of our study is the small sample size. Due to that, we have included unadjusted, slightly higher *P* values to inform about bacterial phyla, classes, and genera and about microbial and human proteins which appeared to change but did not reach statistical significance. These features could be considered in future studies with larger sample sizes or more exhaustive CF metabolic characterization.

Our study provides an extended comparative analysis with robust statistical support that could optimize the use of both approaches for gut microbiota research. Metaproteomics provides information on composition and functionality, as well as data on host-microbiome interactions (41). Its strength is the identification and quantification of *Actinobacteria* and certain classes of *Firmicutes*, whereas its main limitation is the lack of comparability of alpha diversity indices. Taking all the results into account, both techniques detected an aberrant microbiota in infants with CF during their first year of life, dominated by the enrichment of *R. gnavus* within a human inflammatory environment.

## MATERIALS AND METHODS

**Participants, sample collection, and medical records.** Subjects included in this study had CF diagnosed by neonatal screening, sweat chloride test, and mutation sequencing and were recruited during their first month of life, when the first fecal sample was collected, between January 2018 and January 2019. The infants were attended at 3 Spanish CF reference hospitals, geographically distant. Ethical approval for the study was granted by the Ramón y Cajal Ethics Committee in 2017 with (file number 163/17), and all infants' parents gave informed consent. Subjects were excluded if they were born after that date or had fewer than 3 collected samples or if important clinical data were missing. Eight infants with CF were finally followed up, and the relevant data, including feeding habits, delivery mode, and CF mutations, are shown in Table 5. Data related to antibiotic and other treatments are summarized in "Antibiotic treatments" in the supplemental material. Fecal samples were sequentially collected from the diaper of each infant during the first year of life over routinary medical checkups. Samples were stored at −80℃ in 2 aliquots for subsequent amplicon sequencing and metaproteomics analysis.

**DNA extraction and 16S rRNA amplicon sequencing.** Our study followed guidelines for the Strengthening The Organization and Reporting of Microbiome Studies (STORMS) reporting ("STORM checklist" in the supplemental material) (42). Total DNA from the fecal samples was obtained with the QiaAMP kit (Qiagen, Germany), and further MiSeq 2 × 300-bp paired-end (Illumina) 16S rRNA sequencing of the V3 and V4 regions (43) was performed at the Central Unit for Translational Genomics Support (Ramon y Cajal Health Research Institute). The sequencing data analysis was performed using the Qiime pipeline (44), which includes DADA2 for sequence quality filtering, and the SILVA 138 database (released December 2019) was used for taxonomical assignment, discharging those samples with fewer than 1,000 reads (n = 2). Nucleotide sequences were deposited in the National Center for Biotechnology Information's Sequence Read Archive (SRA) repository, BioProject identifier (ID) PRJNA719717.

**Metaproteomics.** Samples were prepared using differential centrifugation, and the microbial pellet was processed by sonication according to a previously published protocol. In brief, 0.1 to 0.3 g of feces was suspended in 10 mL of phosphate-buffered saline (PBS) and mixed in a tube rotator for 45 min at 4℃, as previously published (45, 46). The samples were centrifuged at 500 × *g* for 5 min, and the supernatant was collected in a 50-mL tube. Ten milliliters of PBS was added again, repeating the process twice, and the 3 supernatants (~30 mL) were centrifuged at 11,000 × *g*. Microbial pellets were suspended in 500 µL of lysis buffer (4% sodium dodecyl sulfate [SDS], 50 mM Tris-HCl [pH 8.0]) and heated for 10 min at 95℃. Four sonication cycles (30 s with a 1-min interval on ice) were performed, with an amplitude of 40%. Silica beads were then added

(0.3 g) to each sample, and 5 rounds of bead beating were performed (30 s with a 5-min interval on ice) at a speed of 6.5 ms$^{-1}$. Centrifugation at 14,000 × $g$ was performed to remove the beads and cell debris. To remove the SDS, proteins were precipitated using methanol-chloroform and suspended in 8 M urea for in-solution trypsin digestion. Lastly, the peptides were quantified in a Qubit fluorimeter (Thermo Scientific), and 1 $\mu$g of peptides was loaded for reverse-phase nano-liquid chromatography electrospray ionization tandem mass spectrometric analysis on an EASY-nLC 1000 system (Proxeon) coupled to a Q-Exactive HF mass spectrometer (Thermo Scientific). Peptides were loaded on-line onto an Acclaim PepMap 100 trapping column (75 $\mu$m [inside diameter] by 20 mm, 3 $\mu$m, C$_{18}$ resin with 100-Å pore size; Thermo Scientific) using buffer A (0.1% formic acid) and then separated on a C$_{18}$ resin analytical column (75 $\mu$m [inside diameter] by 500 mm, 2 $\mu$m, 100-Å pore size; EASY-Spray column; Thermo Scientific). A 240-min gradient from 2% to 40% buffer B (0.1% formic acid in 100% acetonitrile) in buffer A was performed to separate the peptides.

Data were obtained by data-dependent acquisition in positive mode. From each MS scan (between 350 and 2,000 Da), the 15 most intense precursors (charges between 2+ and 5+) were selected for their high collision energy dissociation fragmentation, with a dynamic exclusion of 10 s and a normalized collision energy of 20, and the corresponding MS/MS spectra were acquired. Peptides were eluted using a 240-min gradient. The MS proteomics raw data have been submitted to the ProteomeXchange Consortium (http://www.proteomexchange.org) via the Proteomics Identifications Database partner repository with the database identifier PXD029284 (47). For data processing, we employed MetaLab software, which provides a human gut microbial database (12, 48). A human database downloaded from UniProt DB (http://www.uniprot.org) (49), restricted to human taxonomy (downloaded on 18 February 2020 with 74,451 sequences), was also used to identify human proteins. For peptide and protein identification, the FDR was set to 0.01. Only taxa identified with at least 2 peptides were considered and were manually filtered to eliminate human peptides. The sum of the intensities of all the distinctive peptides assigned to a taxon was used as the relative abundance of that taxon. A taxon-function analysis was also performed, using taxonomic information from the enrichment analysis from the iMetaLab platform (http://shiny.imetalab.ca/) (50).

**Graphs and statistical analysis.** The data were trimmed by selecting only the taxa identified by both methods (shared taxa), solving annotation incongruences. First, we evaluated the dichotomous dependent variable, "taxa presence," by a McNemar test with continuity correction at the phylum, class, and genus levels, considering in each sample any amplicon sequence variant (of at least 2 peptides) per taxon. Significance was set to a $P$ value of <0.05. Second, a differential abundance analysis was performed with taxa detected in more than 50% of the samples by at least one of the approaches. Compositional abundance of each phylum, class, and genus was normalized per sample, dividing by total bacterial read counts assigned to the specific taxonomical level and by total sum of LFQ intensity detected at that bacterial taxonomical level for AS and metaproteomics, respectively. Alpha diversity metrics (Shannon index and species richness) were calculated with estimate_richness function from the phyloseq package (version 1.40.0) of RStudio using genus abundances (LFQ or read counts). Package ggpurb (version 0.4.0) was used for plotting paired data. GraphPad Prism (version 9.1.1.225) was used for the functional statistical analysis of the microbial and human proteins.

Nonparametric Wilcoxon signed-rank paired tests were performed to compare the populations' mean ranks (individual taxa and alpha diversity metrics), pairing by sample ID. Significance values in Wilcoxon tests were adjusted via a Benjamini-Hochberg (BH) FDR of 0.1 for AS and metaproteomics abundances.

The CF infant microbiota through the first months of life was profiled using the splinectomeR package (version 0.1.0) in RStudio (version 4.2.1). Differences in the top taxon relative abundances over the time series detected by each method were evaluated with the permuspliner function (999 permutations). Likewise, each taxon was investigated individually to detect any divergence between methods at a particular time point along the time course with the slidingspliner function. Finally, we investigated the time effect for the top taxon relative abundance trends along the time series detected by each method separately using the trendyspliner function. The same approach was used to evaluate differences between methods for alpha diversity metrics. Significance values were adjusted by an FDR of 0.1.

Significant differences were verified with linear models adjusting for age, birth feeding (maternal only/formula), cystic fibrosis transmembrane conductance regulator (CFTR) mutation type (high risk/low risk), and respiratory exacerbations (yes/no), which were considered if one or more episodes were suffered during the sample collection.

**Data availability.** The sequence data from this study are deposited in the NCBI's SRA repository, BioProject ID PRJNA719717. The proteomics data set from this paper has been deposited in the ProteomeXchange Consortium via the PRIDE partner repository with the data set identifier PXD029284.

## SUPPLEMENTAL MATERIAL

Supplemental material is available online only.
**SUPPLEMENTAL FILE 1**, XLSX file, 0.01 MB.
**SUPPLEMENTAL FILE 2**, XLSX file, 0.1 MB.
**SUPPLEMENTAL FILE 3**, XLSX file, 0.1 MB.
**SUPPLEMENTAL FILE 4**, XLSX file, 0.2 MB.
**SUPPLEMENTAL FILE 5**, XLSX file, 0.02 MB.
**SUPPLEMENTAL FILE 6**, XLSX file, 0.03 MB.
**SUPPLEMENTAL FILE 7**, XLSX file, 0.01 MB.

## ACKNOWLEDGMENTS

We thank the families of the children for their participation and all the health personnel of the CF units, and we thank Marta Cobo for technical assistance.

R.D.C. is the recipient of a Vertex Pharmaceuticals grant. The other authors declare no conflicts of interest related to this work.

This work was supported by the Instituto de Salud Carlos III, cofunded by the European Union (PI17/00115 and PI20/00164 to R.D.C.), REIPI 307 (RD16/0016/0011) actions, cofinanced by the European Development Regional Fund "A way to achieve Europe" (ERDF), Vertex Pharmaceuticals. C.S. has grant support from "Fundación Mutua Madrileña" in Atenas Internacional (ATH) 017 call to R.D.C. (AP165902017). Proteomic experiments were supported by projects RTI2018-094004-B-100 and InGEMICS-CM B2017/BMD3691 via funds to C.G.

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
