## [Reviewer comments · Microbiology Spectrum]

Microbiology Spectrum

Statistical evaluation of metaproteomics and 16s rRNA amplicon sequencing techniques for the study of the gut microbiota establishment of infants with cystic fibrosis

Claudia Saralegui, Carmen García-Durán, Eduardo Romeu, María Luisa Hernández-Sánchez, Ainhize Maruri, Natalia Bastón-Paz, Adeliada Lamas, Saioa Vicente, Estela Pérez-Ruiz, Isabel Delgado, Carmen Luna-Paredes, Juan de Dios Caballero, Javier Zamora, Lucía Monteoliva, Concha Gil, and Rosa del Campo

Corresponding Author(s): Rosa del Campo, University Hospital Ramon y Cajal

Review Timeline:

Submission Date:	April 21, 2022
Editorial Decision:	July 5, 2022
Revision Received:	August 24, 2022
Editorial Decision:	September 1, 2022
Revision Received:	September 7, 2022
Editorial Decision:	September 12, 2022
Revision Received:	September 20, 2022
Accepted:	September 23, 2022

Editor: Joanna Goldberg

Reviewer(s): The reviewers have opted to remain anonymous.

Transaction Report:

DOI: <https://doi.org/10.1128/spectrum.01466-22>

July 5, 2022

Dr. Rosa del Campo
University Hospital Ramon y Cajal
Microbiology
Ctra. Colmenar Km 9,1
Madrid 28034
Spain

Re: Spectrum01466-22 (Statistical evaluation of metaproteomics and 16s rRNA amplicon sequencing techniques for the study of the gut microbiota establishment of infants with cystic fibrosis)

Dear Dr. Rosa del Campo:

Thank you for submitting your manuscript to Microbiology Spectrum. This manuscript has now been reviewed by two experts in the field. You will see that each of them had a large number of MAJOR concerns that need to be addressed before this paper can be considered ready for publication.

Link Not Available

Sincerely,

Joanna Goldberg

Journals Department
Reviewer comments:

Reviewer #1 (Comments for the Author):

In this report, the investigators use 16S amplicon sequencing and proteomics to study the intestinal microbiota of infants with CF. The sought to assess the agreement between these two methods using a small cohort of 8 infants with CF, and a total of 28 fecal samples. For most taxa they saw good agreement between the methods, with some clear exceptions. This work confirmed some previous findings regarding early stage intestinal microbiota in this population (altered diversity) but could also extend findings via the use of proteomics (particularly the ability to identify specific bacterial proteins including virulence factors).

The strengths of this study are the combine approaches and the ability to detect bacterial proteins. Limitations/concerns include: (i) the rather small cohort size, (ii) the large differences observed for some taxa between the two methods, (iii) the discrepancy with SDI measures compared to other studies. This observation alone could easily be explained by the small n , but the concern is that the authors still try to make broad conclusions with the limited data presented, and (iv) the somewhat mixed message of the abstract and Intro with the rest of the manuscript. That is, I came into the manuscript thinking I was going to learn about the relative value of 16S vs. proteomics, but much of the Results and Discussion focus on conclusions about changes in taxa/proteins over time in this small sample set. These are really two different questions - this aspect could be addressed by re-working the front part of the manuscript to make it clear that you have two goals - but it does need to be clear that the latter goal has a number of important caveats. The statistical approaches should be validated with some with expertise in these tools.

Specific Comments.

1. Given that previous research supports that the main drivers of microbiome-host interactions are not proteins (ie short chain fatty acids, glycolipids), it would have been beneficial to submit samples for additional analysis using methods that can determine concentrations of these secreted factors to complement the proteomics. Alternatively, a rationale for proteomics vs. metabolomics would have been helpful.
2. In the abstract they say the CF gut microbiota is characterized by poor alpha diversity - I assume this compared to health controls?
3. Comments regarding the Introduction:
 - a. The authors state that the activity of gut microbiota has an impact on nutritional status but gut microbiota's impact spans more than nutritional status.
 - b. The authors state that high intake for antibiotics for exacerbations but these are taken for bacterial infections that are usually overlapping with exacerbations.
4. Line 77-78. "Overrepresentation has been found for certain genera, whereas others, mainly Bifidobacterium, are consistently underrepresented". I would argue that the two most consistent differences seen in young children with CF is the reduced Bacteroides and increased E. coli (and other proteobacteria), at least in the two largest cohorts published to date.
5. Sources are missing (line 86-88), or are you referring to their own data?
6. In trying to understand the statistical test they used, I could not find how Bland-Altman test is referenced/applied. There is a references to another paper that applied this test, but then that paper just argues the famous adage "absence of evidence is not evidence of absence".
7. Small sample size: They only had 28 fecal samples from CF infants and argue that they are studying sequential establishment of the gut microbiota - is 8 infants enough here? No power analysis was performed?
8. Lines 80-86. How accurate is metaproteomics in terms of relative abundance. Overall, this section is written as if 16S amplicon sequencing is the worst approach ever, and with one exception, metaproteomics is the best. Let us be realistic in that each approach has strengths and weaknesses and the presentation, in my opinion, should be a bit more balanced. And I do have to say, no one ever talks about how expensive "meta" studies are and how that limits study size (just my pet peeve!).
9. Table 1. I have not seen "Eutocic" used as a descriptor for birth.
10. Line 121: "Overestimation". As stated, the authors seem to be implying that one method is "correct" and the other method is "over estimating" or "under estimating" taxa. I think this should be reworded to simply state there is a statistically significant difference in the taxa predicted by each method. This same issue should be addressed in the abstract.
11. Line 126: No explanation is given as to what is driving the differences in taxa abundances between methods.
12. When comparing initial vs established microbiome samples - the established samples range from ages 7 months to 12 months, this is a huge difference as other studies show that alpha diversity can change quite a lot in the first year of life, even from month to month.
13. There's a lot of variation in the CF mutation genotype and birth feeding. Not clear if these are accounted for.
14. Related to the point above: Why didn't you use sample 9 for infant 3 as an established microbiome sample?
15. Figure 3 focuses on differences between taxa, which is useful for the reader. One concern is the large differences in some cases (up to 80%). However, from these data it is difficult to get a sense of the taxa that do not change. I think it would be very helpful to plot the data as a transition plate at the phylum and genus level - with statistically significant difference indicated on the plot. (See PMID 31209076, Figure 5 as an example).
16. Line 130. Can you clarify what you mean by "evolved ($n=8$) samples" here. Do you mean the sampling at a time after birth? If so, I would not use the term "evolved" - this has a specific meaning - perhaps "post-birth sample" or something similar. Pls replace throughout.
17. Line 132. "the alpha diversity indexes were higher in the initial samples," This finding is at odds with other data, but do keep in mind that this is a small cohort, and the effects are only statistically significant at the Class levels. Perhaps it might be better to say "the alpha diversity indexes trended towards higher in the initial samples," or something similar. The fact that CF samples showed higher diversity than healthy control is also unexpected.
18. Table 2 - not sure how to read this table. What do they mean that the databases showed incongruencies in how they classified taxa? Other than correcting for this, how is it relevant to comparing the two methods?
19. Table 3 figure description has typo "were or were not"
20. Lines 135-137. "To decipher the relevant abundance fluctuations in the evolved samples, only the taxonomic traits with the greatest positive or negative variations were considered after discarding the initial abundance." Please clarify the rationale for

this approach. It is not clear to me.

21. Related to the point above. The manuscript purports to compare the relative similarity/differences in the utility of 16S and metaproteomics approaches in analyzing these stool samples. At the same time, the authors are using this small data set (and including this information in the abstract) to make conclusions about the microbiome over the first ~1 year of life. The problem with the latter is that some basic findings found in many other studies around SDI early/late and CF/healthy are at odds with much of the literature. I think this point should be mentioned in the abstract, and that it is likely the consequence of the small n of the study. The small n does not invalidate the basic comparison, but I am concerned that making statements about which genera change without also clearly highlighting the caveats of the study (small size, discrepancy with other datasets) has the potential to muddy the literature.

22. Figure 2, for abundance of protein extraction - what is the rest of the white space?? Also, if we know 16s can't get down to species level then plot at genus level

23. For the section titled "Functional assignment by metaproteomics", this is again off base with what the title and abstract focus on - here you are not doing a comparison with 16S, but rather drawing conclusions from 8/16 samples (depending on the analysis). In this context, is it surprising that general COG class (energy, transport, etc.) are not different??

24. I am skeptical of figure 4, newborn vs early CF classification of samples as the early CF samples are spread out. Figure 4B, guessing that left is amplicon sequencing and right is metaproteomics but that is not completely clear in the figure as it is with A and C. No explanation is given for why there are differences between methods.

25. In Figure 5B you use "Early CF" - this works better than "evolved".

26. Figure 5c is not a histogram but it says so in the text below figure

27. Line 159-161. "Bifidobacterium spp. was the most abundant taxa and was also responsible for most of the bacterial functions in the initial samples (Figure 5B)." It would be surprising if this were not the case, correct?

28. Lines 162-64. "A significant enrichment of *R. gnavus* proteins was observed from baseline to evolved times (2 vs. 69 proteins), most of them related to translational processes, such as amino acid and carbohydrate metabolism (Figure 5C)." Two points. (i) As *Ruminococcus* was the only bug to increase significantly in your samples, one would also expect the proteins to go up. As is the case with point #12, I would say that this is more a control/confirmation than a result. (ii). I am not sure I agree that changes in "amino acid and carbohydrate metabolism" should be linked to "translational processes". Are they eating the aa or making them? And carbohydrate metabolism could also be linked to capsule/EPS, correct?

29. The section starting on line 171 is a laundry list of proteins - just direct the reader to a Table. These could just be abundant proteins because there is really no comparison group. Also, again you are trying to drive conclusions based on 8 samples. I would be cautious here. Also, these data have little to do with the comparison of the two methods which I thought was the focus of the paper.

30. The Discussion should be significantly shortened and start with a caveat of the sample size. Basically, the lines between 270-277 are the most relevant, though I do think "extended" is a bit of an over statement. Much of the rest of the Discussion makes pretty broad claims based on a small data set. The authors could even consider a combined Results/Discussion with a short summary at the end.

Reviewer #2 (Comments for the Author):

In their manuscript, Saralegui et al. investigate the agreement between 16S rRNA amplicon sequencing and metaproteomics by using both techniques to profile the gut microbiota of infants with cystic fibrosis. The work described is of interest, but I have some concerns that would need to be addressed:

- The calculation of a difference in relative abundances (Figure 3) is not convincing. Is the total microbial signal in the different samples comparable? What happens if, for the normalization of the metaproteomics data, the relative abundance is calculated using the sum of the microbial signal instead of the total signals? The rationale behind the calculation performed (subtracting metaproteomics abundances to amplicon sequencing ones - lines 365-366) should be better explained. Moreover, considering the heterogeneity between subjects (Lines 367-371), the methods used to assess the taxonomical differences (Figure 4) are even less convincing.

- Figure 3, it would be interesting to know from which samples the discordances are coming. Is there a connection between the observed differences and the clinical data (Additional file 1)?

- Lines 135-142, these results are probably due to the small number of samples analyzed. Reason for which the relevance of these fluctuations (both those detected by amplicon sequencing and metaproteomics) remains to be validated. These aspects should be discussed by the authors.

- Line 324, 1 ug is a pretty high amount of peptides. Have the authors checked the carryover and used blank samples in between? In addition, no replicates are mentioned, meaning that the abundances are calculated on single measures, this could be a critical aspect for low abundant taxa.

- The potential impact of the protocol followed for extracting the metaproteomes from the fecal samples and, more specifically, of the incubation of the samples several times for 45 min at 4{degree sign}C on the functional analysis should be discussed.

- Lines 207-209, the confidence of the metaproteomics analysis at the species level should be discussed. Is there enough signal to be confident of identification at the species level (line 231)? A comparison at the genus level could be more appropriate and equally informative.

- From line 270, The authors should assess and validate (statistical analysis) the added value of combining data from the two techniques to profile CF infants' microbiota.

Minor points

- It is unclear why in Fig 1 is indicated 30 samples if the analysis of 28 samples is described in the text.

- Lines 284-286 and 290-292 are a repetition. Moreover, the number of the study should be mentioned.

- Line 354, the R package should be mentioned.

Staff Comments:

Preparing Revision Guidelines

Please return the manuscript within 60 days; if you cannot complete the modification within this time period, please contact me. If you do not wish to modify the manuscript and prefer to submit it to another journal, please notify me of your decision immediately so that the manuscript may be formally withdrawn from consideration by Microbiology Spectrum.

In their manuscript, Saralegui et al. investigate the agreement between 16S rRNA amplicon sequencing and metaproteomics by using both techniques to profile the gut microbiota of infants with cystic fibrosis. The work described is of interest, but I have some concerns that would need to be addressed, in my opinion:

- The calculation of a difference in relative abundances (Figure 3) is not convincing. Is the total microbial signal in the different samples comparable? What happens if, for the normalization of the metaproteomics data, the relative abundance is calculated using the sum of the microbial signal instead of the total signals? The rationale behind the calculation performed (subtracting metaproteomics abundances to amplicon sequencing ones - lines 365-366) should be better explained. Moreover, considering the heterogeneity between subjects (Lines 367-371), the methods used to assess the taxonomical differences (Figure 4) are even less convincing.
- Figure 3, it would be interesting to know from which samples the discordances are coming. Is there a connection between the observed differences and the clinical data (Additional file 1)?
- Lines 135-142, these results are probably due to the small number of samples analyzed. Reason for which the relevance of these fluctuations (both those detected by amplicon sequencing and metaproteomics) remains to be validated. These aspects should be discussed by the authors.
- Line 324, 1 ug is a pretty high amount of peptides. Have the authors checked the carryover and used blank samples in between? In addition, no replicates are mentioned, meaning that the abundances are calculated on single measures, this could be a critical aspect for low abundant taxa.
- The potential impact of the protocol followed for extracting the metaproteomes from the fecal samples and, more specifically, of the incubation of the samples several times for 45 min at 4°C on the functional analysis should be discussed.
- Lines 207-209, the confidence of the metaproteomics analysis at the species level should be discussed. Is there enough signal to be confident of identification at the species level (line 231)? A comparison at the genus level could be more appropriate and equally informative.
- From line 270, The authors should assess and validate (statistical analysis) the added value of combining data from the two techniques to profile CF infants' microbiota.

Minor points

- It is unclear why in Fig 1 is indicated 30 samples if the analysis of 28 samples is described in the text.
- Lines 284-286 and 290-292 are a repetition. Moreover, the number of the study should be mentioned.
- Line 354, the R package should be mentioned.

Madrid 15 of August of 2022

Dear editor,

We appreciate the review of our manuscript which help us to improve its scientific quality. After considering these comments we have decided to include all samples in the analysis of fecal microbiota changes during the first months of life of children with CF. The questions have been answered one by one and changes to the manuscript have been marked in the track changes document.

REVIEWER #1: In this report, the investigators use 16S amplicon sequencing and proteomics to study the intestinal microbiota of infants with CF. The sough to assess the agreement between these two methods using a small cohort of 8 infants with CF, and a total of 28 fecal samples. For most taxa they saw good agreement between the methods, with some clear exceptions. This work confirmed some previous findings regarding early stage intestinal microbiota in this population (altered diversity) but could also extend findings via the use of proteomics (particularly the ability to identify specific bacterial proteins including virulence factors).

The strengths of this study are the combine approaches and the ability to detect bacterial proteins. Limitations/concerns include: (i) the rather small cohort size, (ii) the large differences observed for some taxa between the two methods, (iii) the discrepancy with SDI measures compared to other studies. This observation alone could easily be explained by the small n, but the concern is that the authors still try to make broad conclusions with the limited data presented, and (iv) the somewhat mixed message of the abstract and Intro with the rest of the manuscript. That is, I came into the manuscript thinking I was going to learn about the relative value of 16S vs. proteomics, but much of the Results and Discussion focus on conclusions about changes in taxa/proteins over time in this small sample set. These are really two different questions - this aspect could be addressed by re-working the front part of the manuscript to make it clear that you have two goals - but it does need to be clear that the latter goal has a number of important caveats. The statistical approaches should be validated with some with expertise in these tools.

Answer: We agree with the reviewer's comments, and we will try to answer one by one the limitations that have been pointed out.

(i) the rather small size of the cohort: It is really a small cohort, but considering the nature of CF, a rare disease, it has been impossible for us to recruit more children, although we have appealed to several hospitals in our country. Our initial intention

was to recruit 50 children, but after 3 years we decided to stop recruiting. Another determining factor was the lack of clinical data or samples; unfortunately, on some occasions we had samples but no data, or the other way around. This also limited the number of children included.

(ii) the large differences observed for some taxa between the two methods: It is not clear to us that this is a limitation, but rather the result of our research and the main objective of the work in which we intended to compare the results obtained by proteomics and 16S rDNA amplicon sequencing.

(iii) the discrepancy with SDI measurements compared to other studies. This observation alone could easily be explained by the small n, but the concern is that the authors are still trying to make broad conclusions with the limited data presented: We agree on this point, consequently we have softened the discussion message about SDIs on the proteomic results.

(iv) the somewhat mixed message of the abstract and Intro with the rest of the manuscript. That is, I went into the manuscript thinking I was going to learn about the relative value of 16S versus proteomics, but much of the Results and Discussion focus on conclusions about changes in taxa/proteins over time in this small set of samples. In reality, these are two different questions - this aspect could be addressed by reworking the initial part of the manuscript to make it clear that there are two objectives - but it does need to be made clear that the latter objective has a number of important caveats. Statistical approaches should be validated by people experienced with these tools: Our main objective was to perform a statistically comparison of the results obtained with both methodologies, but we consider it appropriate to also reflect the results that we obtained since there are not too many studies of microbiota implantation in CF. We agree with the reviewer, and therefore introduce the second objective to make it clear in the abstract.

One of the authors of the paper is a recognized researcher in the field of Clinical Biostatistics (Javier Zamora, ORCID 0000-0003-4901-588X). He has made important contributions to this work, in addition to supervising the entire manuscript, introducing P-adjustment by FDR.

Specific Comments.

Question 1: Given that previous research supports that the main drivers of microbiome-host interactions are not proteins (ie short chain fatty acids, glycolipids), it would have been beneficial to submit samples for additional analysis using methods that can determine concentrations of these secreted factors to complement the proteomics. Alternatively, a rationale for proteomics vs. metabolomics would have been helpful.

Answer: We agree with the reviewer that an integrative analysis including all types of omics is the best option. In this case, the limited amount of stool we can collect from the diaper does not allow us to apply any other technique. We are now implementing the fatty acid short chain study in another set of infants, but our results are still ongoing.

Question 2: In the abstract they say the CF gut microbiota is characterized by poor alpha diversity - I assume this compared to health controls?

Answer: Yes, our criteria was the paper published by Antosca et al., and we have included the appropriate reference and explanation in the new version.

Question 3: Comments regarding the Introduction: a. The authors state that the activity of gut microbiota has an impact on nutritional status but gut microbiota's impact spans more than nutritional status.

Answer: Totally agree, but in the field of CF we are referring specifically to maldigestion of fats and malabsorption of nutrients, in addition to alterations of the intestinal microbiota. All this means that patients often present nutritional problems with a considerable impact on the progression of the disease.

Question: b. The authors state that high intake for antibiotics for exacerbations but these are taken for bacterial infections that are usually overlapping with exacerbations.

Answer: The reviewer is right; we have modified this sentence in the new version.

Question 4: Line 77-78. "Overrepresentation has been found for certain genera, whereas others, mainly *Bifidobacterium*, are consistently underrepresented". I would argue that the two most consistent differences seen in young children with CF is the reduced *Bacteroides* and increased *E. coli* (and other Proteobacteria), at least in the two largest cohorts published to date.

Answer: We only detailing the limitations of amplicon sequencing technique.

Question 5: Sources are missing (line 86-88), or are you referring to their own data?

Answer: Sorry for the missing reference, it has been included in the new version.

Question 6: In trying to understand the statistical test they used, I could not find how Bland-Altman test is referenced/applied. There is a reference to another paper that applied this test, but then that paper just argues the famous adage "absence of evidence is not evidence of absence".

Answer: Bland-Altman is only a comparative representation, not a method per se. In the new version we have removed this figure and included a new figure based on the dot plot, as suggested by the reviewer, in which the Wilcoxon test has been used.

Question 7: Small sample size: They only had 28 fecal samples from CF infants and argue that they are studying sequential establishment of the gut microbiota - is 8 infants enough here? No power analysis was performed.

Answer: Totally agree! Our goal was to recruit 50 children and we incorporated other hospitals to increase recruitment as CF is a rare disease. After 3 years of recruitment, many of the children did not have all the sequential samples, in others we did not have all the clinical data and finally we could only have all the requirements complete in 8 children. We could have chosen other older patients as well, but we found it much more interesting to perform the study with these complicated samples, which were small in quantity and the microbiota was very changeable, since we intended to evaluate the discriminatory power of genomics and proteomics as a proof of concept.

Question 8: Lines 80-86. How accurate is metaproteomics in terms of relative abundance. Overall, this section is written as if 16S amplicon sequencing is the worst approach ever, and with one exception, metaproteomics is the best. Let us be realistic in that each approach has strengths and weaknesses and the presentation, in my opinion, should be a bit more balanced. And I do have to say, no one ever talks about how expensive "meta" studies are and how that limits study size (just my pet peeve!).

Answer: In comparisons one of the methods is always considered the gold standard, and in this case, we decided that amplicon sequencing was the gold standard, but the reviewer is right, and we have softened the message in the new version. There are metaproteomics studies where they process samples to account for peptide signal

difference and then work with relative abundances, e.g., Riffle 2017, 10.3390/proteomes6010002. We have also introduced a sentence with the economic aspect, and we fully agree with the reviewer. Until now proteomics was considerably more expensive, but prices have become much tighter (€150 per sample), although obviously far from genomics prices (€50 per sample).

Question 9: Table 1. I have not seen "Eutocic" used as a descriptor for birth.

Answer: Sorry for the mistake that has been corrected in the new version.

Question 10: Line 121: "Overestimation". As stated, the authors seem to be implying that one method is "correct" and the other method is "over estimating" or "under estimating" taxa. I think this should be reworded to simply state there is a statistically significant difference in the taxa predicted by each method. This same issue should be addressed in the abstract.

Answer: We fully agree with the reviewer and the changes have been properly introduced in the new version.

Question 11: Line 126: No explanation is given as to what is driving the differences in taxa abundances between methods.

Answer: The particularity of each methodology is detailed in the introduction section, and a specific paragraph has now been added in the discussion.

Question 12: When comparing initial vs established microbiome samples - the established samples range from ages 7 months to 12 months, this is a huge difference as other studies show that alpha diversity can change quite a lot in the first year of life, even from month to month.

Answer: I We consider this point to be critical and agree with the reviewer. Consequently, we have reanalyzed the data again by including all 28 available samples in a linear model, increasing the statistical power in the analysis.

Question 13: There's a lot of variation in the CF mutation genotype and birth feeding. Not clear if these are accounted for.

Answer: Following the new criteria for classifying CF mutations into high and low risk according to their clinical manifestations as it has been published (Rueda-Nieto et al.,

2022), our population is homogeneous with high-risk mutations in 7 infants, while only patient 4 is considered low risk.

Question 14: Related to the point above: Why didn't you use sample 9 for infant 3 as an established microbiome sample?

Answer: All available samples have been included in the new analysis.

Question 15: Figure 3 focuses on differences between taxa, which is useful for the reader. One concern is the large differences in some cases (up to 80%). However, from these data it is difficult to get a sense of the taxa that do not change. I think it would be very helpful to plot the data as a transition plate at the phylum and genus level - with statistically significant difference indicated on the plot. (See PMID 31209076, Figure 5 as an example).

Answer: Following the reviewer's recommendations, we have changed the figure, and we appreciate the suggestion, as this new figure is much more informative.

Question 16: Line 130. Can you clarify what you mean by "evolved (n=8) samples" here. Do you mean the sampling at a time after birth? If so, I would not use the term "evolved" - this has a specific meaning - perhaps "post-birth sample" or something similar. Pls replace throughout.

Answer: Sorry for the confusion of the message, these terms have been changed to newborn and early CF status to make it easier to understand.

Question 17: Line 132. "the alpha diversity indexes were higher in the initial samples," This finding is at odds with other data, but do keep in mind that this is a small cohort, and the effects are only statistically significant at the Class levels. Perhaps it might be better to say "the alpha diversity indexes trended towards higher in the initial samples," or something similar. The fact that CF samples showed higher diversity than healthy control is also unexpected.

Answer: We have not included the proteomics diversity indices in the new version of the manuscript to avoid confusing messages.

Question 18: Table 2 - not sure how to read this table. What do they mean that the databases showed incongruencies in how they classified taxa? Other than correcting for this, how is it relevant to comparing the two methods?

Answer: Table 2 has been modified to improve their understanding.

Question 19. Table 3 figure description has typo "were or were not"

Answer: The mistake has been corrected in the new version.

Question 20: Lines 135-137. "To decipher the relevant abundance fluctuations in the evolved samples, only the taxonomic traits with the greatest positive or negative variations were considered after discarding the initial abundance." Please clarify the rationale for this approach. It is not clear to me.

Answer: The message has been modified in the new version to avoid misunderstandings.

Question 21: Related to the point above. The manuscript purports to compare the relative similarity/differences in the utility of 16S and metaproteomics approaches in analyzing these stool samples. At the same time, the authors are using this small data set (and including this information in the abstract) to make conclusions about the microbiome over the first ~1 year of life. The problem with the latter is that some basic findings found in many other studies around SDI early/late and CF/healthy are at odds with much of the literature. I think this point should be mentioned in the abstract, and that it is likely the consequence of the small n of the study. The small n does not invalidate the basic comparison, but I am concerned that making statements about which genera change without also clearly highlighting the caveats of the study (small size, discrepancy with other datasets) has the potential to muddy the literature.

Answer: In full agreement with the reviewer, we have modified the manuscript according to this recommendation.

Question 22: Figure 2, for abundance of protein extraction - what is the rest of the white space?? Also, if we know 16s can't get down to species level then plot at genus level.

Answer: The white space corresponded to taxon that are correctly identified by both methods, and the species by 16S has been removed from the figure.

Question 23: For the section titled "Functional assignment by metaproteomics", this is again off base with what the title and abstract focus on - here you are not doing a comparison with 16S, but rather drawing conclusions from 8/16 samples (depending

on the analysis). In this context, is it surprising that general COG class (energy, transport, etc.) are not different??

Answer: This information is obtained from metaproteomics, and although we are aware of the limitations, we propose to keep this section because it reflects a relevant approach to this technique.

Question 24: I am skeptical of figure 4, newborn vs early CF classification of samples as the early CF samples are spread out. Figure 4B, guessing that left is amplicon sequencing and right is metaproteomics but that is not completely clear in the figure as it is with A and C. No explanation is given for why there are differences between methods.

Answer: Figure 4 has been modified according to the recommendations.

Question 25: In Figure 5B you use "Early CF" - this works better than "evolved".

Answer: These terms have been changed to newborn and early CF status to make it easier to understand.

Question 26: Figure 5c is not a histogram but it says so in the text below figure.

Answer: It has been corrected in the new version.

Question 27: Line 159-161. "Bifidobacterium spp. was the most abundant taxa and was also responsible for most of the bacterial functions in the initial samples (Figure 5B)." It would be surprising if this were not the case, correct?

Answer: The referee is right; we have modified the sentence in the new version.

Question 28: Lines 162-64. "A significant enrichment of *R. gnavus* proteins was observed from baseline to evolved times (2 vs. 69 proteins), most of them related to translational processes, such as amino acid and carbohydrate metabolism (Figure 5C)." Two points. (i) As *Ruminococcus* was the only bug to increase significantly in your samples, one would also expect the proteins to go up. As is the case with point #12, I would say that this is more a control/confirmation than a result.

Answer: We agree with the reviewer and have emphasized in the text that not only is the population increasing, but it is also a functionally active population, since cell density does not always correspond to increase a particular functionality.

Question: (ii). I am not sure I agree that changes in "amino acid and carbohydrate metabolism" should be linked to "translational processes". Are they eating the aa or making them? And carbohydrate metabolism could also be linked to capsule/EPS, correct?

Answer: The reviewer is right, and we have modified this sentence in the new version.

Question: There is a grammatical error in the sentence, instead it should read: "A significant enrichment of *R. gnavus* proteins was observed from baseline to evolutionary times (2 vs. 69 proteins), most of them related to translational processes and amino acid and carbohydrate metabolisms (Figure 5C)."

Answer: We have corrected it in the text.

Question 29: The section starting on line 171 is a laundry list of proteins - just direct the reader to a Table. These could just be abundant proteins because there is really no comparison group. Also, again you are trying to drive conclusions based on 8 samples. I would be cautious here. Also, these data have little to do with the comparison of the two methods which I thought was the focus of the paper.

Answer: we agree with this assessment and have introduced Table 5.

Question 30: The Discussion should be significantly shortened and start with a caveat of the sample size. Basically, the lines between 270-277 are the most relevant, though I do think "extended" is a bit of an over statement. Much of the rest of the Discussion makes pretty broad claims based on a small data set. The authors could even consider a combined Results/Discussion with a short summary at the end.

Answer: The discussion has been modified as proposed by the reviewer, but we are unable to reduce it without taking away relevant information

REVIEWER #2: In their manuscript, Saralegui et al. investigate the agreement between 16S rRNA amplicon sequencing and metaproteomics by using both techniques to profile the gut microbiota of infants with cystic fibrosis. The work described is of interest, but I have some concerns that would need to be addressed:

Question: The calculation of a difference in relative abundances (Figure 3) is not convincing. Is the total microbial signal in the different samples comparable? What happens if, for the normalization of the metaproteomics data, the relative abundance is calculated using the sum of the microbial signal instead of the total signals? The rationale behind the calculation performed (subtracting metaproteomics abundances to amplicon sequencing ones - lines 365-366) should be better explained. Moreover, considering the heterogeneity between subjects (Lines 367-371), the methods used to assess the taxonomical differences (Figure 4) are even less convincing.

Answer: We follow the recommendation of Riffle et al (10.3390/proteomes6010002): *“Spectral counts are then normalized using one of various methods [20,21,22,23]. For example, the Normalized Spectrum Abundance Factor (NSAF) [22] adjusts the spectral counts based on the total number of proteins identified in the sample and the respective lengths of those proteins. Finally, the normalized values, functional annotations and taxonomic assignments for the predicted protein lists are used to ascribe relative abundances to functions and taxa and these abundances are compared between samples”.*

Question: Figure 3, it would be interesting to know from which samples the discordances are coming. Is there a connection between the observed differences and the clinical data (Additional file 1)?

Answer: We have modified figure 3 in the new version, but no correlation with the clinical data was detected.

Question: Lines 135-142, these results are probably due to the small number of samples analyzed. Reason for which the relevance of these fluctuations (both those detected by amplicon sequencing and metaproteomics) remains to be validated. These aspects should be discussed by the authors.

Answer: We agree that our main limitation is the small sample size, and this comment has been repeatedly introduced in the new version.

Question: Line 324, 1 ug is a pretty high amount of peptides. Have the authors checked the carryover and used blank samples in between? In addition, no replicates are mentioned, meaning that the abundances are calculated on single measures, this could be a critical aspect for low abundant taxa.

Answer: In our experience, 1 ug of peptides is not very much, even in other works they use a higher amount, such as 4 ug (Zhang X, Chen W, Ning Z, Mayne J, Mack D, Stintzi

A, Tian R, Figeys D. Deep Metaproteomics Approach for the Study of Human Microbiomes. *Anal Chem.* 2017 Sep 5;89(17):9407-9415. doi: 10.1021/acs.analchem.7b02224). Of course, a blank is always included between samples to detect possible contamination, but biological replicates were limited by the small amount of sample available.

Question: The potential impact of the protocol followed for extracting the metaproteomes from the fecal samples and, more specifically, of the incubation of the samples several times for 45 min at 4°C on the functional analysis should be discussed.

Answer: This step is used in several metaproteomics studies as a method to gently disperse the initial sample for example:

- Long et al. Metaproteomics characterizes human gut microbiome function in colorectal cancer. *NPJ Biofilms Microbiomes.* 2020 Mar 24;6(1):14. doi: 10.1038/s41522-020-0123-4.
- Abbondio et al. Fecal metaproteomic analysis reveals unique changes of the gut microbiome functions after consumption of sourdough *Carasau* bread. *Front Microbiol.* 2019 Jul 30;10:1733. doi: 10.3389/fmicb.2019.01733.

Furthermore, when comparing the use of this 45 min wetting step or the use of 5 min vortexing, we did not observe any significant difference between the use of one or the other in terms of the functional profile of the samples (Garcia-Duran et al., 2021). The protein patterns observed on a Coomassie gel were also similar:

- Protocol A: 45 min wetting step
- Protocol D: 5 min vortexing.

Question: Lines 207-209, the confidence of the metaproteomics analysis at the species level should be discussed. Is there enough signal to be confident of identification at the species level (line 231)? A comparison at the genus level could be more appropriate and equally informative.

Answer: We have introduced these items in the discussion section.

Question: From line 270, The authors should assess and validate (statistical analysis) the added value of combining data from the two techniques to profile CF infants' microbiota.

Answer: We have followed previously published recommendations (Subramanian et al. Multi-omics data integration, interpretation, and its application. *Bioinform Biol Insights*. 2020 Jan 31;14:1177932219899051. doi: 10.1177/1177932219899051).

Minor points

Question: It is unclear why in Fig 1 is indicated 30 samples if the analysis of 28 samples is described in the text.

Answer: Sorry for the mistake, it has been corrected in the new version.

Question: Lines 284-286 and 290-292 are a repetition. Moreover, the number of the study should be mentioned.

Answer: Modified according to it has been suggested.

Question: Line 354, the R package should be mentioned.

Answer: This information has been included in the new version

September 1, 2022

Dr. Rosa del Campo
University Hospital Ramon y Cajal
Microbiology
Ctra. Colmenar Km 9,1
Madrid 28034
Spain

Re: Spectrum01466-22R1 (Statistical evaluation of metaproteomics and 16s rRNA amplicon sequencing techniques for the study of the gut microbiota establishment of infants with cystic fibrosis)

Dear Dr. Rosa del Campo:

Thank you for submitting your manuscript to Microbiology Spectrum. As you will see your paper is very close to acceptance. Please modify the manuscript as recommended by the 2 reviewers (below). As these revisions are quite minor, I expect that you should be able to turn in the revised paper in less than 30 days, if not sooner.

When submitting the revised version of your paper, please provide (1) point-by-point responses to the issues raised by the reviewers as file type "Response to Reviewers," not in your cover letter, and (2) a PDF file that indicates the changes from the original submission (by highlighting or underlining the changes) as file type "Marked Up Manuscript - For Review Only". Please use this link to submit your revised manuscript. Detailed instructions on submitting your revised paper are below.

Link Not Available

Sincerely,

Joanna Goldberg

Reviewer comments:

Reviewer #1 (Comments for the Author):

The authors have made a strong effort to respond to the previous critiques. The abstract is much improved and aligns with what is presented in the manuscript. They also reworked the text to soften some of the conclusions so that the conclusions also align with the data presented. This work is a nice contribution to the literature. I have a few minor comments on the revised manuscript that I hope will clarify the presentation.

1. In the abstract, (lines 46-47) where the authors use the terms overestimation and underestimation - this implies that one method is giving a correct value and one is not, but you have nothing to determine which is which. This same issue came up in the first version of the manuscript. I would replace these terms with "higher predicted relative abundance" and "lower predicted relative abundance", respectively.
2. Line 60 - delete "massive". The techniques are not massive, the data sets they generate are.
3. Line 67, delete ", which is a novel result"
4. Line 69: replace "to" with "and the enrichment of" - you have no evidence for a direct link between this microbe and the observed inflammatory response.

Reviewer #2 (Comments for the Author):

Revisions have substantially improved the manuscript.

As a further comment, I would like to point out that I disagree with the general statement in lines 65-66 (but also 154-155). Although standardization is still needed, an increasing number of reports show that metaproteomic data allow for the profiling of microbiota composition and alpha diversity analysis.

There are a few small English errors in the text, for example line 49 should be "characterized by a significant", line 239 "discrepancies were found", etc.

Preparing Revision Guidelines

Please return the manuscript within 60 days; if you cannot complete the modification within this time period, please contact me. If you do not wish to modify the manuscript and prefer to submit it to another journal, please notify me of your decision immediately so that the manuscript may be formally withdrawn from consideration by Microbiology Spectrum.

Madrid 1 of September of 2022

Dear editor and reviewers,

Thanks for your positive comments, the new suggestions have been included.

REVIEWER #1:

The authors have made a strong effort to respond to the previous critiques. The abstract is much improved and aligns with what is presented in the manuscript. They also reworked the text to soften some of the conclusions so that the conclusions also align with the data presented. This work is a nice contribution to the literature. I have a few minor comments on the revised manuscript that I hope will clarify the presentation.

Question 1: In the abstract, (lines 46-47) where the authors use the terms overestimation and underestimation - this implies that one method is giving a correct value and one is not, but you have nothing to determine which is which. This same issue came up in the first version of the manuscript. I would replace these terms with "higher predicted relative abundance" and "lower predicted relative abundance", respectively.

Question 2: Line 60 - delete "massive". The techniques are not massive, the data sets they generate are.

Question 3: Line 67, delete ", which is a novel result"

Question 4: Line 69: replace "to" with "and the enrichment of" - you have no evidence for a direct link between this microbe and the observed inflammatory response.

Answer: All suggested changes has been made in the new version.

REVIEWER #2: Revisions have substantially improved the manuscript.

Question 1: As a further comment, I would like to point out that I disagree with the general statement in lines 65-66 (but also 154-155). Although standardization is still needed, an increasing number of reports show that metaproteomic data allow for the profiling of microbiota composition and alpha diversity analysis.

Answer: The message has been smoothed in the new version, but for lines 144-145 we cannot change the result that proteomics failed to detect singletons.

Question 2: There are a few small English errors in the text, for example line 49 should be "characterized by a significant", line 239 "discrepancies were found", etc.

Answer: Sorry for the mistakes, it has been corrected.

September 12, 2022

Dr. Rosa del Campo
University Hospital Ramon y Cajal
Microbiology
Ctra. Colmenar Km 9,1
Madrid 28034
Spain

Re: Spectrum01466-22R2 (Statistical evaluation of metaproteomics and 16s rRNA amplicon sequencing techniques for the study of the gut microbiota establishment of infants with cystic fibrosis)

Dear Dr. Rosa del Campo:

Thank you for submitting your revised manuscript to Microbiology Spectrum.

The changes you made to the text seem to be appropriate and adequately answer the reviewers' concerns, but when I look at the Tables and Figures I now find problems.....

There is highlighted text in Table 1, 2, 3, and 4. I'm not sure whether is left over from a prior version, but please look this over.

Also, in Figure 5, there seems to be an extra panel with "Evolution of the Shanon (spelled wrong) alpha diversity index". Again, I don't know where this came from.

Once you figure out the issues here, please submit a revised manuscript with "Response to Editor's Comments" document. As these revisions are quite minor, I expect that you should be able to turn in the revised paper in less than 30 days, if not sooner.

Detailed instructions on submitting your revised paper are below.

Link Not Available

Sincerely,

Joanna Goldberg

Reviewer comments:

Preparing Revision Guidelines

- Point-by-point responses to the issues raised by the editor in a file named "Response to Editor's Comments", NOT IN YOUR COVER LETTER.
- Upload a compare copy of the manuscript (without figures) as a "Marked-Up Manuscript" file.
- Each figure must be uploaded as a separate file, and any multipanel figures must be assembled into one file.
- Manuscript: A .DOC version of the revised manuscript

- Figures: Editable, high-resolution, individual figure files are required at revision, TIFF or EPS files are preferred

Please return the manuscript within 60 days; if you cannot complete the modification within this time period, please contact me. If you do not wish to modify the manuscript and prefer to submit it to another journal, please notify me of your decision immediately so that the manuscript may be formally withdrawn from consideration by Microbiology Spectrum.

Madrid 15 of September of 2022

Dear editor,

We are sorry for the problems of working with several versions, we apologize for this, we hope that in this version everything is already correct. Figure 5 has been adequately corrected.

Sincerely,

Rosa del Campo